

# CHASE-PL Climate Projection dataset over Poland–Bias adjustment of EURO-CORDEX simulations

Abdelkader Mezghani[1], Andreas Dobler[1], Jan Erik Haugen[1], Rasmus Eduard Benestad[1], Kajsa Maria Parding[1], Mikołaj Piniewski[2,3], Ignacy Kardel[2], and Zbigniew W. Kundzewicz[3]

[1]Norwegian Meteorological Institute, PO.BOX. 43 0313 Blindern, Norway
[2]Department of Hydraulic Engineering, Warsaw University of Life Sciences, Nowoursynowska 166, 02-787 Warsaw, Poland
[3]Potsdam Institute for Climate Impact Research, Telegrafenberg, 14473 Potsdam, Germany

*Correspondence to:* Abdelkader Mezghani (abdelkaderm@met.no)

**Abstract.** The CHASE-PL (Climate change impact assessment for selected sectors in Poland) Climate Projections - Bias Corrected Daily Precipitation and Temperature dataset 5 km (CPLCP-GDPT5) consists of projected daily minimum and maximum air temperatures and precipitation totals of nine EURO-CORDEX regional climate model outputs bias corrected and downscaled to $5 \times 5$ km grid. Simulations of one historical period (1971-2000) and two future horizons (2021-2050 and 2071-2100)

assuming two Representative Concentration Pathways (RCP4.5 and RCP8.5) were produced. We used the quantile mapping method and corrected any systematic bias in these simulations before assessing the changes in annual and seasonal means of precipitation and temperature over Poland. Projected changes estimated from the multi-model ensemble mean showed that annual means of temperature are expected to increase constantly by 1 °C until 2021-2050 and by 2 °C until 2071-2100 assuming the RCP4.5, which is accelerating assuming the RCP8.5 scenario and can reach up to almost 4 °C by 2071–2100. Similarly to

temperature, projected changes in regional annual means of precipitation are expected to increase by 6 % to 10 % and by 8 % to 16 % for the two future horizons and RCPs, respectively. Similarly, individual model simulations also exhibited warmer and wetter conditions on an annual scale, showing an intensification of the magnitude of the change at the end of the 21st century. The same applied for projected changes in seasonal means of temperature showing a higher winter warming rate by up to 0.5 °C compared to the other seasons. However, projected changes in seasonal means of precipitation by the individual models

largely differ and are sometimes inconsistent exhibiting spatial variations which depends on the selected season, location, future horizon and RCP. The overall range of the 90% confidence interval predicted by the ensemble of multi-model simulations was found to likely vary between -7 % and +40 %, expected to occur in summer assuming the RCP4.5 scenarios and in winter assuming the RCP8.5 scenario, respectively, at the end of the 21st century. Finally, this high-resolution bias-corrected product can serve as a basis for climate change impact and adaptation studies for many sectors over Poland. CPLCP-GDPT5 dataset is

publicly available at http://dx.doi.org/10.4121/uuid:e940ec1a-71a0-449e-bbe3-29217f2ba31d.

## 1   Introduction

Regional climate change projections for all terrestrial regions of the globe within the time line of the Fifth Assessment Report (AR5) and beyond have been made available for climate researchers in the framework of the CORDEX initiative. Within this



initiative, a large ensemble of high resolution regional climate projections including Europe (EURO-CORDEX, the European branch of the CORDEX initiative) have been made available to provide climate simulations for use in climate change impact, adaptation, and mitigation studies (Giorgi and Lionello, 2008). Although most of the simulations are run on a high grid resolution, systematic biases in the regional climate models are remaining due to errors related to i) imperfect model representation

of the physical processes or phenomena and ii) to the parametrization and incorrect initialization of the models. Thus, even by using the highest-resolution available regional climate models still require some adjustments (Christensen et al., 2008; Muerth et al., 2013). Therefore, bias correction methods continue to be used in impact studies e.g. hydrology (e.g., Chen et al., 2013; Teutschbein and Seibert, 2012), agronomy (e.g., Ines and Hansen, 2006), ecology, and more recently by climate services (e.g., Sorteberg et al., 2014) to reduce systematic bias in (regional or global) climate models.

Traditionally, the bias correction method ensures equal mean values between the corrected simulations and observations (e.g., Déqué et al., 2007), hence, explicitly addressing only one aspect of the statistical properties of the simulated data. More advanced methods consider the whole distribution of a weather variable to be adjusted, including extremes so that it matches the distribution of the observations (e.g., Themeßl et al., 2010; Berg et al., 2012; Lafon et al., 2013).

Recent studies have compared different RCM bias correction methods. Themeßl et al. (2010) evaluated seven bias correction
methods used to correct modelled precipitation by the regional climate model MM5 using forcings from ERA40 reanalysis. They concluded that quantile mapping outperform all methods considered, especially at high quantiles. Berg et al. (2012) applied three bias correction methods to correct the mean and variance of precipitation and temperature modelled by the RCM COSMO-CLM driven by the ECHAM5-MPIOM GCM over entire Germany and its near surrounding areas modelled at 7 km resolution and validated against 30 years of 1 km gridded observations data (1971–2000). They found that the method
corrects not only means but also higher moments. Gudmundsson et al. (2012) confirmed that non-parametric methods such as quantile mapping are more suitable in reducing systematic errors in model data. They compared eleven bias correction methods used to correct precipitation modelled by RCM HIRHAM forced with the ERA40 reanalysis data and pointed out that non-parametric methods performed the best in reducing systematic errors, followed by parametric transformations with three or more free parameters, and lowest rank for the distribution derived transformations. Teutschbein and Seibert (2012) applied six
bias correction methods to correct eleven different RCM-simulated temperature and precipitation series, and pointed out that all methods were able to preserve the mean, however, other statistical properties were degraded. Lafon et al. (2013) applied four distribution-based bias correction methods to correct precipitation modelled by the regional climate model HadRM3-PPEUK driven by the global climate model HadCM3 over seven catchments in Great Britain. They pointed out that gamma-based quantile mapping offers the best combination of accuracy when evaluated on the first four order moments (mean, standard
deviation, skewness and kurtosis). Sorteberg et al. (2014) tested six distribution-based bias correction methods. They pointed out that all evaluated methods perform reasonably well in i) reproducing statistical properties of the observations including high order moments and quantiles and ii) preserving the climate change signal.

Bias correction methods can also be categorized in parametric and non-parametric methods. In the parametric methods the distribution of the data is assumed to be known. For instance, it is well known that the probability distribution of the daily
temperature values follow a normal distribution (Buishand and Brandsma, 1997), whereas, the exponential (Benestad et al.,



2005) and Gamma distributions (Buishand and Brandsma, 2001) are often used to model the intensity of daily precipitation. Likewise, the Bernoulli and geometric distributions are often used to model the probability distribution of the occurrence of daily precipitation (frequency) and the number of consecutive dry/wet days, respectively (Buishand and Beckmann, 2000). On the other hand, the non-parametric methods are applied without prior assumptions about the distribution of the data (Lanzante, 1996), hence, they are more attractive for many applications including those based on bias correction. Another advantage is that non-parametric methods are more suitable in reducing systematic errors in model data (Gudmundsson et al., 2012).

Among existing methods, the 'Quantile Mapping' method has shown a good performance in reproducing not only the mean and the standard deviation but also other statistical properties such as quantiles (Gutjahr and Heinemann, 2013). As the method belongs to the non-parametric family, it does not require a prior knowledge of the theoretical distribution of the weather variable which makes it very attractive as it is easy to implement, in addition, to its simple and non-parametric configuration (Gudmundsson et al., 2012).

However, the EQM has few limitations. It is particularly sensitive to the choice and the length of the calibration time period to make a reliable estimation not affected by the data sample problems Fowler and Kilsby (2007). Thus, it requires a reference dataset to adjust the modelled data so that it matches the observations (Lafon et al., 2013). It is sometimes difficult to apply this method to different climatic conditions as unobserved values may lay outside the range of those in the calibration time period (Themeßl et al., 2010) (i.e. values in the tail of the distribution). Another issue can be related to the misrepresentation of the (physical) link between weather variables that can be altered especially if it is applied on each climate variable separately. For instance, in most hydrological applications the dependency between daily precipitation and temperature may affect the discharge Haerter et al. (2011).

Few studies related to projections of climate change have been dedicated to Poland. For instance, climate projections originating from the ENSEMBLES project (Linden and Mitchell, 2009) were the basis to investigate the impact of climate change on various sectors (agriculture, water resources and health) in Europe. Szwed et al. (2010) assessed six regional climate model simulations under SRES A2 scenario and pointed out unfavourable changes in Polish climate such as an increased frequency of extreme events, reduced crop yields, and increased summer water budget deficit. In the 'KLIMADA' project, the ENSEMBLES projections were additionally bias-adjusted(http://klimada.mos.gov.pl/en/) within the framework of the Polish National Adaptation Strategy to Climate Change (NAS 2020) to estimate changes in climate variables and indices in two future horizons, the near future (2021-2050) and the far future (2071-2100). The outcomes of the latter project showed significant upward trend in temperature and uncertain precipitation increases in the median of winter changes and slight decreases in summer. Piotrowski and Jędruszkiewicz (2013) assessed the spatial variability in winter temperature over Poland for the near future (2021–2050) based on three regional climate models under the SRES A1B scenario.

There has been also few studies carried out for Poland based on the newest generation of climate model simulations (i.e. the fifth generation of the Coupled Model Inter-comparison Project (CMIP5) and the European domain of the Coordinated Downscaling Experiment Initiative (EURO-CORDEX)). Romanowicz et al. (2016) used bias-adjusted modelled temperature and precipitation (seven GCM-RCM combinations from the EURO-CORDEX initiative over ten Polish catchments) and found that projections following the RCP4.5 agreed on a precipitation increase of up to 15%, and a warming of up to 2°C by the



end of the 21st century. Pluntke et al. (2016) applied a statistical downscaling model to produce temperature and precipitation projections from two global climate models following three different emission scenarios (RCP2.6, RCP8.5, and SRES-A1B) for the south-western part of Poland and eastern Saxony. They pointed out an acceleration of changes by the end of the 21st century leading to negative consequences for the climatic water balance, particularly under SRES A1B and RCP8.5.

5    The main objective of this paper is to provide an update of climate projections over Poland by adopting the new generation of concentration pathways and recent developments in climate modelling. We hope the dataset provided here will be beneficial for the research community, for instance, impact studies in areas such as hydrology, ecology and agricultural sciences. In this paper we also used a recently made available high resolution gridded observational dataset (CPLFD-GDPT5, see Sect. 2) covering more than 60 years as a reference for the bias correction procedure (Sect. 3.1).

## 10    2    Input datasets

In the present study two types of datasets were used: (1) a Polish high-resolution observational climate datasets used as reference for the bias-correction and (2) a (multi-model) ensemble of regional climate model simulations provided through the EURO-CORDEX experiment.

### 2.1    Polish high resolution observational climate datasets

The gridded daily precipitation and temperature dataset (CPLFD-GDPT5) is used here as reference or pseudo-observational data in the bias adjustment procedure (Berezowski et al., 2016). The dataset consists of a 5 ×5 km gridded product of daily precipitation, minimum air temperature, and maximum air temperature. The spatial extent of the CPLFD-GDPT5 is the union of two intersecting areas: the Vistula and Odra river basins and Poland's territory. It covers the period from 1951 to 2013 (63 years). Berezowski et al. (2016) evaluated the CPLFD-GDPT5 data on reproducing observed Polish climate and concluded that

the new high resolution gridded product show a good consistency with previous products, albeit small differences arose due to the assimilation of new sets of meteorological stations, using different interpolation techniques and performing. Piniewski et al. (2017b) used this dataset as inputs in hydrological modelling of the Vistula and Odra river basins and reported satisfactory model performance in simulating daily discharges in 110 flow gauges. To our knowledge, it is the best currently available climatic dataset that could be used as reference in bias correction in this study. For simplicity, it will be hereafter referred to as

'observations'.

### 2.2    Regional climate model simulations

The regional climate model simulations, referred hereafter to as 'simulations', consist of nine historical simulations spanning the time period from 1949 to 2005 and of 18 model simulations spanning the future time period from 2006 to 2100 provided within the EURO-CORDEX initiative. From these simulations, we extracted daily minimum and maximum temperatures and

precipitation on grid cells belonging to the same spatial domain as the observations i.e. the area of Poland and parts of the Vistula and Odra basins belonging to neighbouring countries. This domain corresponds to the area from 13.1 to 26.1 degrees



**Table 1.** GCM/RCM simulations.

| N | Global Climate Model | | | Regional Climate Model | | Period | |
| | Institute | Model | Run | Institute | Model | From | To |
|---|-----------|-------|-----|-----------|-------|------|-----|
| 1 | CNRM-CERFACS | CNRM-CM5 | r1i1p1 | CLMcom | CCLM4-8-17 | 1950-01-01 | 2100-12-31 |
| 2 | CNRM-CERFACS | CNRM-CM5 | r1i1p1 | SMHI | RCA4 | 1970-01-01 | 2100-12-31 |
| 3 | ICHEC | EC-EARTH | r12i1p1 | CLMcom | CCLM4-8-17 | 1949-12-01 | 2100-12-31 |
| 4 | ICHEC | EC-EARTH | r12i1p1 | SMHI | RCA4 | 1970-01-01 | 2100-12-31 |
| 5 | ICHEC | EC-EARTH | r1i1p1 | KNMI | RACMO22E | 1950-01-01 | 2100-12-31 |
| 6 | ICHEC | EC-EARTH | r3i1p1 | DMI | HIRHAM5 | 1951-01-01 | 2100-12-31 |
| 7 | IPSL | IPSL-CM5A-MR | r1i1p1 | SMHI | RCA4 | 1970-01-01 | 2100-12-31 |
| 8 | MPI-M | MPI-ESM-LR | r1i1p1 | CLMcom | CCLM4-8-17 | 1970-01-01 | 2100-12-31 |
| 9 | MPI-M | MPI-ESM-LR | r1i1p1 | SMHI | RCA4 | 1949-12-01 | 2100-12-31 |

East and 48.6 to 54.9 degrees North. The total number of the grid cells equals $N_g = 23016$ ($168 \times 137$). Selected simulations consisted of the combination of four global climate models (GCM) and four regional climate models following the two representative concentration pathways RCP4.5 and RCP8.5 and are presented in Table 1. We also focused on three common time slices spanning the periods 1971-2000 referred further to as control period and two future horizons 2021-2050 and 2071-2100 referred further to as near and far futures, respectively. As those simulations were made available on different spatial resolutions, an interpolation onto the same $5 \times 5$ km grid as for observations was performed before the bias correction method was applied.

## 3 Data Analyses

### 3.1 Bias correction method

We used the quantile mapping to correct for systematic biases in regional climate model simulations. The quantile mapping tries to find a statistical transformation or function $F$ that maps a simulated variable $y$ such that its new distribution fits closely the distribution of the observed variable $x$. In general, this transformation can be formulated as

$$x = F(y) \tag{1}$$

The non-parametric transformation is then defined as (Piani and Haerter, 2012)

$$x = F^{-1}(G(y)) \tag{2}$$

where $G$ is the cumulative distribution function of $y$ and $F^{-1}$ is the inverse cumulative distribution function corresponding to $x$. The quantile mapping of the simulated time series to the observed ones was performed for each grid cell. Here, the



number of quantiles was set to $N_q = 1000$ and was chosen to be regularly spaced. Two steps were performed. First, RCM corresponding quantiles were taken from the empirical cumulative distribution function based on observations. Second, these estimates were used to perform a quantile mapping. It should be noted that the set-up included a linear interpolation between the fitted transformed values and simulated values lying outside the range of observed values in the training period, hence, were

extrapolated using the correction found for the highest percentile as suggested by Boé et al. (2007). Furthermore, the method included an adjustment of wet-day frequencies for precipitation. The quantile mapping method was additionally applied on each of the four seasons separately to take into account for seasonality in the observations as different seasons may be influenced by different physical processes. Then, the output data were merged to reconstruct a full simulation. As discussed in Sect. 1, the quantile mapping may modify the link between individually post-processed climate variables, however, correcting the present

climate to be closer to the observations has been necessary for most climate change impact studies (Sorteberg et al., 2014). Quantiles of the simulations for the control period (1971–2000) were mapped onto corresponding quantiles in the observations considered as the most recent 63-years reference time period (1951–2013). The transfer functions were then used to correct for the bias in the daily temperature and precipitation simulations defined in Table 1.

## 3.2   Biases in regional climate model simulations

Each of the nine bias corrected historical simulations was evaluated on its ability to reproduce statistical properties of the pseudo-observed or reference dataset. In our case, averaged error values ($e$) over time ($t$) for each grid cell in terms of root mean square errors were considered as a measure of the models' performance. As model errors are often seasonally dependent, the four seasons are treated separately and for a time $t$ (seasonal or annual), the model error is calculated as

$$e_t = s_t - o_t \tag{3}$$

where $s$ and $s$ refer to simulated and observed values. The root mean square error averaged over space is then defined as

$$RMSE = \sqrt{\sum_{i=1}^{N_g} e^2{}_t} \tag{4}$$

where $Ng$ is the total number of grid cells.

    The averaged $RMSE$ informs about the magnitude of the overall deviation between the simulations and pseudo-observations over all Poland, while, the error ($e$) indicates whether there was an over (positive) or under (negative) estimation (bias) of the

simulated values at each grid cell.

    The root mean square error was computed between values of bias corrected and raw monthly sums of precipitation, minimum and maximum daily temperatures and their corresponding observations. The model error was first computed on each grid cell, then mapped across Poland and averaged from the spatial field only, i.e. the RMSE of the temporal means of all grid cells, not at the single grid cells.

Tables 2 to 4 gives the mean and standard deviation of the RMSE derived from the ensemble of model simulations.

    The root mean square error in annual means averaged over all raw simulations (i.e. multi-model ensemble mean) was 15.51±4.39 mm/month for precipitation and 1.09±0.65 °C and 1.64±0.45 °C for minimum and maximum daily temperatures,



**Table 2.** Root mean square errors in annual and seasonal means of monthly sums of precipitation (mm/month). The root mean square errors were computed between historical simulations and observations and averaged over all grid cells.

| N | GCM/RCM simulation | Annual | Winter | Spring | Summer | Autumn |
|---|---|---|---|---|---|---|
| 1 | CNRM-CM5/CCLM4-8-17 | 13.45 | 11.71 | 7.00 | 45.19 | 8.50 |
| 2 | CNRM-CM5/RCA4 | 15.09 | 14.54 | 19.49 | 27.25 | 12.16 |
| 3 | EC-EARTH/CCLM4-8-17 | 12.64 | 12.15 | 8.05 | 28.45 | 10.22 |
| 4 | EC-EARTH/RCA4 | 14.57 | 15.82 | 17.6 | 24.33 | 11.41 |
| 5 | EC-EARTH/RACMO22E | 13.12 | 14.68 | 13.1 | 20.33 | 9.78 |
| 6 | EC-EARTH/HIRHAM5 | 19.4 | 21.71 | 18.4 | 22.67 | 20.93 |
| 7 | IPSL-CM5A-MR/RCA4 | 23.16 | 21.64 | 33.88 | 27.24 | 19.87 |
| 8 | MPI-ESM-LR/CCLM4-8-17 | 8.74 | 13.20 | 8.68 | 13.30 | 9.99 |
| 9 | MPI-ESM-LR/RCA4 | 19.44 | 14.76 | 26.56 | 29.08 | 16.19 |
| Ens. mean* | All | 15.51 | 15.58 | 16.97 | 26.43 | 13.23 |
| Ens. sd* | All | 4.39 | 3.70 | 9.01 | 8.61 | 4.62 |

* Ens. stands for ensemble, sd for standard deviation

respectively (Tables 2 to 4). For seasonal means, the largest error was found in summer precipitation (26.43 ±8.61 mm/month), mainly due to the convection, not well represented in the models. The lowest error was found in the autumn, where precipitation is influenced by continental air masses. The same tendency was additionally found for daily maximum temperature, i.e. large error in the summer compared to the other seasons. However, for minimum daily temperature, the largest error of

1.53 °C was obtained for spring, followed by summer with a slightly lower bias of 1.32 °C. In general, biases in maximum daily temperatures were 0.5 to 1 °C higher than those found in minimum daily temperatures. The lowest precipitation bias was simulated by the regional climate model RCA4 driven by the global model M-MPI-ESM-LR (Simulation 8 in Table 2). Obviously, these biases or model errors might be related to the complexity of the climate system in Poland which has been very difficult to predict, being influenced by air masses from all four directions (Kundzewicz and Matczak, 2012).

Moreover, model errors or biases are often spatially dependant and varied among the simulations and seasons. In our case, all historical simulations showed wet and warm biases as well as dry and cold biases across the region, which were more pronounced in the mountainous areas located in the southern parts of Poland, due to topographical features not well represented in the models (SM 1 to 27). This can also be related to the low observational network density in this region. As we did not intend to perform a thorough comparison between all model simulations, only an example of the bias in the regional climate

model CCLM4-8-17 driven by the CNRM-CM5 global climate model for the historical climate (Simulation 1 in Table 1) is detailed hereafter (Figures 1–3).

For precipitation, seasonal evaluations additionally showed that the relatively high root mean square errors found in the raw data on annual scale (13.45 mm/month) were due to high biases in summer and winter which were 45.9 mm/month and





**Table 3.** Root mean square errors in annual and seasonal means of daily minimum temperatures (°C). The root mean square errors were computed between historical simulations and observations and averaged over all grid cells.

| N | GCM/RCM simulation | Annual | Winter | Spring | Summer | Autumn |
|---|---|---|---|---|---|---|
| 1 | CNRM-CM5/CCLM4-8-17 | 0.68 | 1.34 | 1.03 | 2.29 | 0.76 |
| 2 | CNRM-CM5/RCA4 | 1.13 | 1.65 | 1.94 | 0.73 | 0.86 |
| 3 | EC-EARTH/CCLM4-8-17 | 0.63 | 0.73 | 0.59 | 0.93 | 0.82 |
| 4 | EC-EARTH/RCA4 | 1.43 | 1.21 | 1.6 | 2.1 | 1.27 |
| 5 | EC-EARTH/RACMO22E | 2.66 | 2.16 | 4.02 | 2.11 | 2.54 |
| 6 | EC-EARTH/HIRHAM5 | 0.63 | 0.91 | 0.71 | 0.54 | 0.78 |
| 7 | IPSL-CM5A-MR/RCA4 | 0.95 | 1.21 | 1.77 | 1.59 | 0.77 |
| 8 | MPI-ESM-LR/CCLM4-8-17 | 0.84 | 0.95 | 1.16 | 0.69 | 0.76 |
| 9 | MPI-ESM-LR/RCA4 | 0.82 | 1.45 | 0.92 | 0.88 | 0.94 |
| Ens.mean* | All | 1.09 | 1.29 | 1.53 | 1.32 | 1.06 |
| Ens.std* | All | 0.65 | 0.43 | 1.05 | 0.70 | 0.58 |

\* Ens. stands for ensemble, sd for standard deviation

**Table 4.** Root mean square errors in annual and seasonal means of maximum daily temperatures (°C). Root mean square errors were computed between historical simulations and observations and averaged over all grid cells.

| N | GCM/RCM simulation | Annual | Winter | Spring | Summer | Autumn |
|---|---|---|---|---|---|---|
| 1 | CNRM-CM5/CCLM4-8-17 | 1.51 | 2.37 | 2.79 | 1.6 | 1.6 |
| 2 | CNRM-CM5/RCA4 | 1.26 | 1.57 | 2.81 | 1.14 | 0.57 |
| 3 | EC-EARTH/CCLM4-8-17 | 1.62 | 1.49 | 1.83 | 1.61 | 1.99 |
| 4 | EC-EARTH/RCA4 | 1.81 | 1.12 | 2.47 | 2.77 | 1.3 |
| 5 | EC-EARTH/RACMO22E | 1.91 | 0.86 | 2.76 | 2.66 | 1.82 |
| 6 | EC-EARTH/HIRHAM5 | 2.39 | 1.52 | 2.14 | 3.39 | 2.83 |
| 7 | IPSL-CM5A-MR/RCA4 | 1.73 | 0.72 | 3.3 | 2.8 | 1.03 |
| 8 | MPI-ESM-LR/CCLM4-8-17 | 1.78 | 1.25 | 1.38 | 2.77 | 1.92 |
| 9 | MPI-ESM-LR/RCA4 | 0.76 | 0.77 | 0.94 | 2.14 | 0.63 |
| Mean | All | 1.64 | 1.30 | 2.27 | 2.32 | 1.52 |
| Standard deviation | All | 0.45 | 0.52 | 0.76 | 0.74 | 0.72 |

\* Ens. stands for ensemble, sd for standard deviation

11.71 mm/month, respectively, compared to the transition seasons with relatively low biases (7 mm/month for spring and 8.5 mm/month for the autumn). Moreover, the highest discrepancy of the models was obtained in mountainous areas located in the south (negative bias range of -75—100 mm/month) and the eastern part of the region with a negative bias in the range





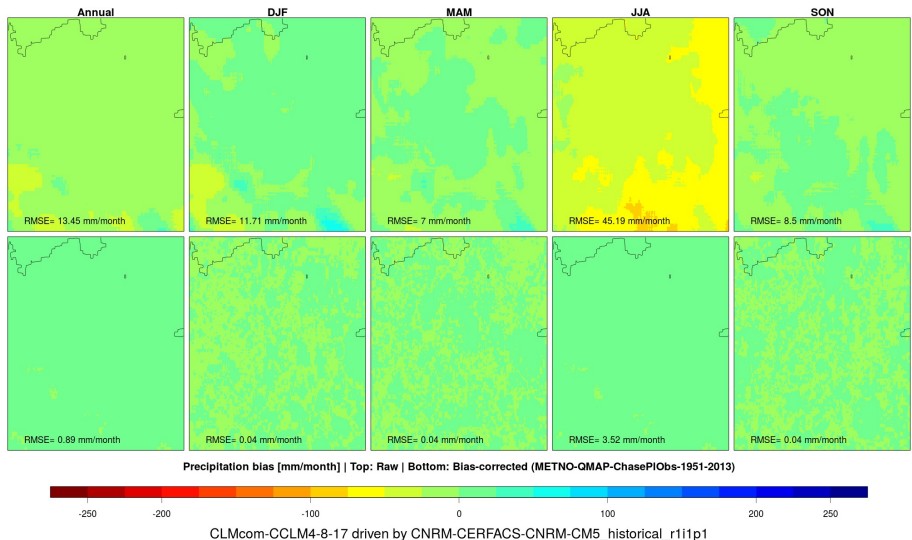

**Figure 1.** Evaluation of the RMSE of annual and seasonal monthly sums of precipitation. Root mean square error maps estimated on the difference between historical simulations (all available years included) and observations (CPLFD-GDPT5) for both raw (top) and bias adjusted (bottom) simulations, respectively. The black line in the upper part of each map indicates the limit between the land and the Baltic sea.

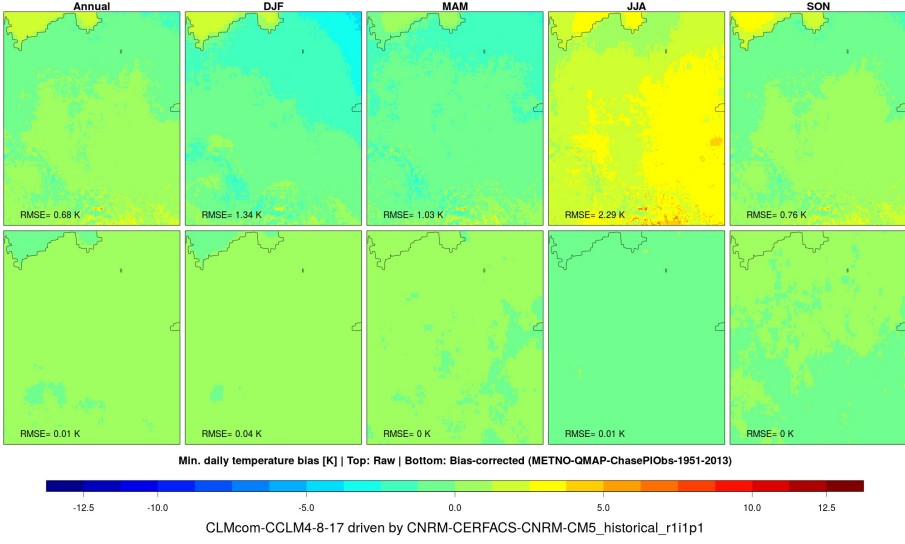

**Figure 2.** Same caption as in Fig. 1 but for daily minimum temperature

of -50—75 mm/month. Obviously the RMSE from the bias adjusted results were very low compared to those obtained from the raw simulations and were mostly close to zero for annual and seasonal means, albeit summer (the summer) where a small





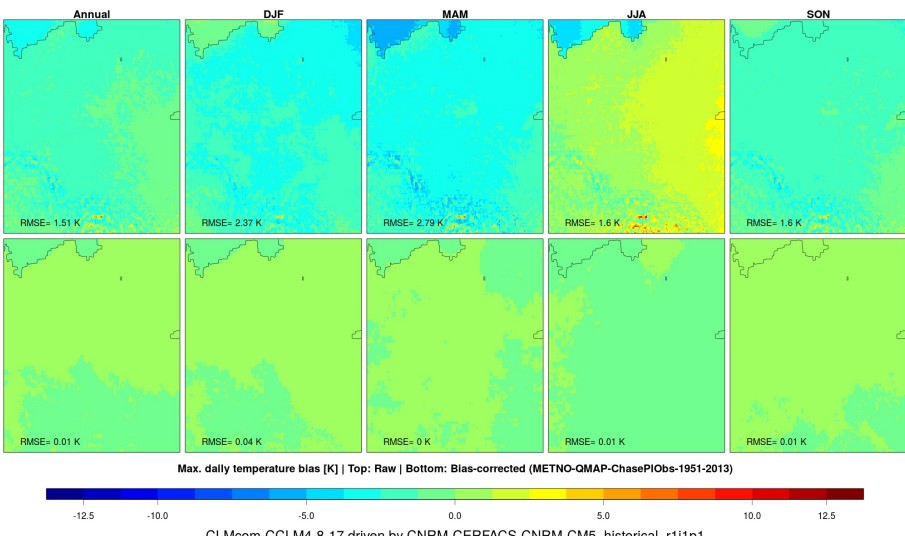

**Figure 3.** Same caption as in Fig. 1 but for daily maximum temperature

bias of 3.52 mm/month persisted in the adjusted precipitation. For minimum temperature, there was an overall cold bias for all seasons except for summer which shows a positive bias in the eastern part of the region and especially in the mountains located in the south. The annual RMSE of the raw data was 1.5 °C. Seasonal evaluations showed higher cold biases for winter and spring of 2.37 °C and 2.79 °C, respectively. A warm bias was found for summer all over the country, which was influenced by

mountains located in the south. For the corrected results, the bias was reduced to almost zero for all seasons and on annual scale. For maximum temperature, the RMSE was slightly lower than for minimum temperature for all seasons except summer, where the RMSE shows relatively high value of 2.29 °C. However, the spatial distribution of the RMSE shows similar pattern to that discussed earlier for minimum temperature. The RMSE based on corrected datasets are all close to zero for both minimum and maximum daily temperatures, however, there was still a spatial structure of the errors. Biases were also removed in corrected

precipitation, albeit, summer precipitation where small biases remained with less than 2 mm/month for all simulations.

### 3.3   Sensitivity to the climate change signal

Although the bias correction significantly improved the quality of the simulations in the trained control time period, it may alter the physical link between climate variables in the model (Ehret et al., 2012), hence, modifying the climate change signal (Teng et al., 2015). We further investigated the influence of the quantile mapping method on the climate change signal. Accordingly,

we mapped the climate change signal in both raw and corrected simulations and focused on the time period 2096-2100 with regard to historical data as we would expect a stronger alteration of the climate signal by the end of the century than earlier. An example of results based on the regional climate model RCA4 driven by the global climate model MPI-ESM-LR is illustrated hereafter (Simulation 9 in Table 1). Figures 4 to 6 suggested that both the magnitude and the spatial distribution of the estimated changes were maintained, hence, not affected by the correction procedure. This demonstrated the reliability of the projected



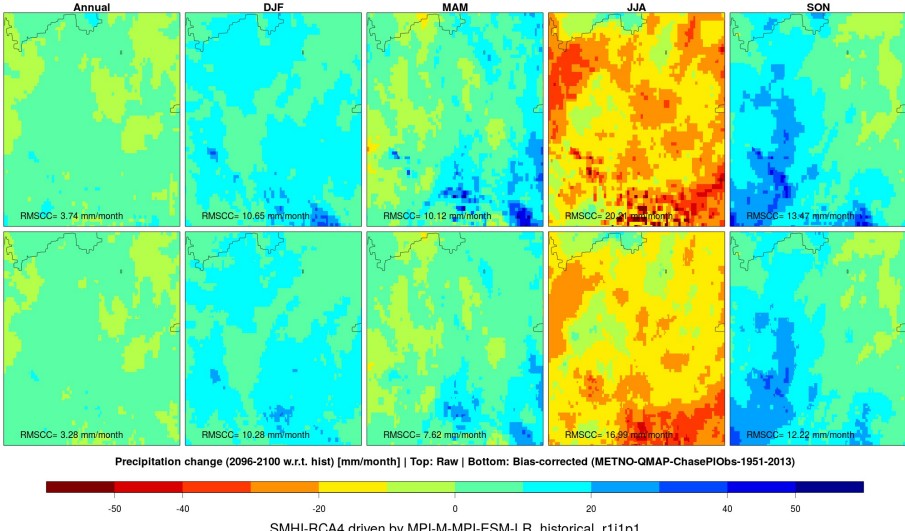

**Figure 4.** Precipitation change signal (mm/month) for simulation 9. Relative changes in future (2096–2100) with regards to historical simulations for both raw (top) and bias adjusted (bottom) data, respectively. The RMSCC indicates the root mean square values of the climate change estimated over the whole spatial domain.

climate changes by corrected regional climate model simulations. One possible explanation could be related to the use of long reference record (1951–2013) on which the calibration was performed i.e. the training distribution has been built on long record encompassing different climate conditions compared to a short reference period that are commonly selected as reference (e.g. 1971–2000 or the new normal 1981–2010). However, few exceptions were found, for instance, the magnitude of the climate change (in terms of root mean square) between historical and future (2096–2100) simulations for both raw and bias adjusted modelled precipitation was reduced by approximately 15 % to 25 % in corrected summer and spring changes, respectively, as opposed to Hagemann et al. (2005), who reported that the impact of the bias correction on the climate change signal may be larger than the signal itself. Overall, the spatial distribution of the climate change signal was, however, consistent in all corrected simulations, i.e. no random effect was introduced after the correction. Similar results were obtained for other regional climate models and even when assuming the RCP8.5 scenario (SM28 to SM54).

## 4  Projected future climate changes in Poland

The dynamical downscaling performed here involved bias-adjusted regional climate model simulations taken from the EURO-CORDEX experiment and corrected against the gridded daily dataset CPLFD-GDPT5 (Berezowski et al., 2016). From these datasets, we calculated climatic changes expressed in terms of relative changes in monthly sums of precipitation (in %) and absolute changes in mean temperature (in °C) with respect to the control period (1971–2000). The mean temperature values were calculated as the average between minimum and maximum temperature values. Although the projections cover small parts

©c Author(s) 2017. CC BY 4.0 License.



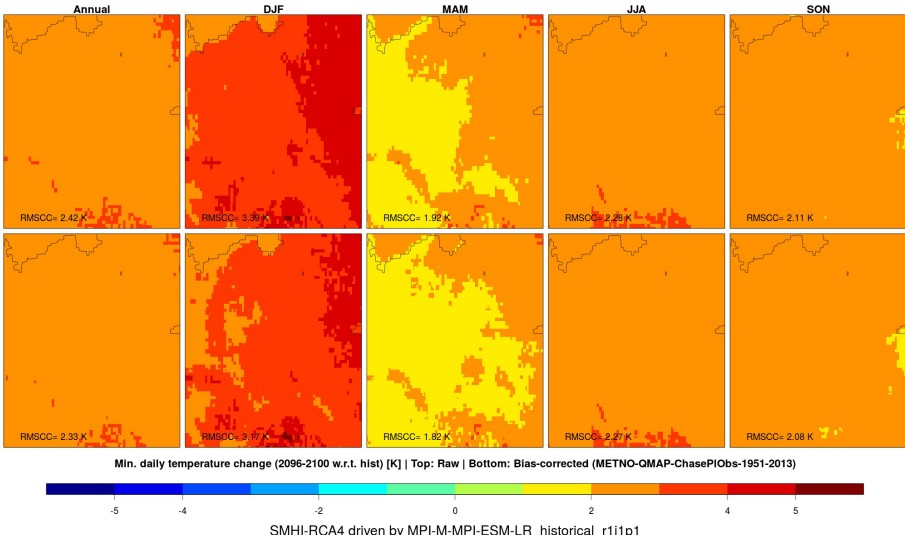

**Figure 5.** Same caption as in Fig. 4 but for absolute changes in minimum temperature (°C).

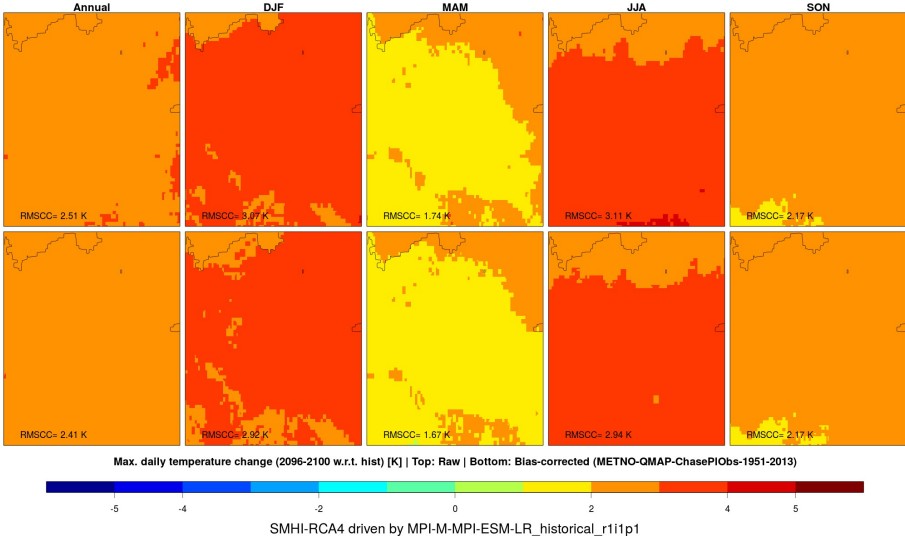

**Figure 6.** Same caption as in Fig. 4 but for absolute changes in maximum temperature (°C).

lying outside Poland, the maps presented here show only changes over Polish territory (Piniewski et al., 2016). We followed a two-fold assessment procedure. First, we evaluated the multi-model ensemble means in projecting changes in annual and seasonal means of monthly sums of precipitation and daily means of temperature (Figures 7 and 8). Second, we focused on projected changes in annual and seasonal means of monthly sums of precipitation and daily minimum and maximum

5    temperature by individual model simulations (See the supplementary material).



**Table 5.** Summary of changes in projected multi-model ensemble seasonal and annual regional means of temperature (a, in °C) and precipitation (b, in %) by the near (2021-2050) and far (2071-2100) futures assuming both the RCP4.5 and RCP8.5. Values between brackets indicate the 5th and 95th percentiles of the projected ensembles and, hence, represent the 90% confidence interval of the mean estimates from the multi-model ensemble.

| Scenario/Horizon | DJF | MAM | JJA | SON | Annual |
|---|---|---|---|---|---|
| **(a) Temperature changes** | | | | | |
| the RCP4.5 by 2021-2050 | +1.2 | +1.0 | +1.0 | +1.1 | +1.1 |
| | [+0.4,+1.9] | [+0.6,+1.7] | [+0.7,+1.4] | [+0.6,+1.6] | [+0.7,+1.4] |
| RCP8.5 by 2021-2050 | +1.6 | +1.3 | +1.1 | +1.3 | +1.3 |
| | [+0.5,+2.5] | [+0.9,+2] | [+0.7,+1.3] | [+0.6,+1.8] | [+0.8,+1.8] |
| the RCP4.5 by 2071-2100 | +2.5 | +2.0 | +1.7 | +1.8 | +2 |
| | [+1.1, +3.3] | [+1.1,+2.8] | [+1.3,+2.3] | [+1.4,+2.4] | [+1.4,+2.5] |
| RCP8.5 by 2071-2100 | +4.5 | +3.2 | +3.1 | +3.5 | +3.6 |
| | [+3.8, +5.3] | [+2.5,+4.0] | [+2.5,+3.9] | [+2.7,+4.2] | [+3.0,+4.1] |
| | | | | | |
| **(b) Precipitation changes** | | | | | |
| RCP4.5 by 2021-2050 | +8.4 | +7.6 | +3.8 | +5.6 | +5.9 |
| | [+2, +17] | [+2,+14] | [-2, +9] | [-2,+14] | [+4,+9] |
| RCP8.5 by 2021-2050 | +13.2 | +10.5 | +4.7 | +6.8 | +8.0 |
| | [+6,+22] | [+0.5,+22.9] | [+0.2,+11] | [+1,+15] | [+5,+11] |
| RCP4.5 by 2071-2100 | +18.4 | +14.8 | +4.0 | +6.5 | +9.7 |
| | [+12,+27] | [+7,+23] | [-7,+12] | [0,+12] | [+6,+13] |
| RCP8.5 by 2071-2100 | +26.8 | +26.4 | +5.2 | +13.1 | +15.7 |
| | [+18,+35] | [+16,+39] | [-5,+15] | [-1,+25] | [+9,+23] |

## 4.1 Changes in the multi-model ensemble mean

### 4.1.1 Projected temperature

Results suggested an ubiquitous warming over Poland in future (Table 5, a). The annual mean temperature over Poland is expected to increase by approximately 1.1 °C for the period 2021-2050 and by 2 °C for the period 2070-2100 following the

5 RCP4.5, with very low spatial variability (spatial standard deviation 0.2 °C, Fig. 7). On a seasonal basis, the highest change is expected to occur in winter (2.5 °C), followed by spring (2.0 °C), and autumn (1.8 °C), and the lowest in summer (1.7 °C). Similarly to the annual mean changes in temperature, the seasonal changes in temperature also exhibit low spatial variability with a span of approximately 0.1 °C (See supplementary material).




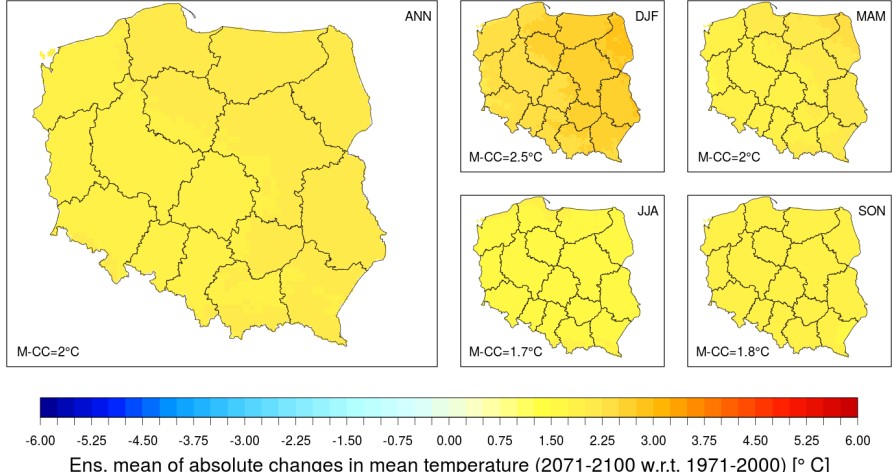

**Figure 7.** Projected temperature changes (°C) for the far future (2071 - 2100) assuming the RCP4.5 scenario. Maps show annual (left large panel) and seasonal (right small panels) changes in the multi-model ensemble mean of absolute temperature with regards to the control period (1971-2000). The legend 'M-CC' means the areal mean change estimated from the gridded data.

The warming rate is accelerating assuming the RCP8.5 emission scenario and when considering the far future time horizon (Fig. 8). As in the RCP4.5, the warming is expected to be the highest during the winter season and is likely to be 4.5 °C across the region and exhibits a clear north-east to south-west gradient. This is in line with (Piotrowski and Jędruszkiewicz, 2013) who pointed out that it is mainly attributed to an increase in the frequency of cyclonic circulation types. In summer, the strongest warming is likely to occur in the mountainous regions in the south where temperatures may rise by as much as 3 ° by 2071-2100.

### 4.1.2 Projected precipitation changes

Projections showed that Poland is expected to get more precipitation in the future for all seasons (Table 5,b). In general, the projections based on the two scenarios show similar changes for the near future. But for the far future, the RCP8.5 high emission scenario projects a significantly stronger increase.

Assuming the intermediate emission scenario RCP4.5, the expected annual mean precipitation increase (averaged over Poland) is approximately 6% by the near future (2021-2050) and 10% by the far future (2071-2100). On a seasonal basis, the highest rates are expected in winter (+8% by 2021-2050 and +18% by 2071-2100) and spring (+8% by 2021-2050 and +15% by 2071-2100), while the smallest changes are expected to occur in summer (+4% for both future time periods) and autumn (+6%/+7%, regardless the future period). Those projected changes are in line with Romanowicz et al. (2016) who





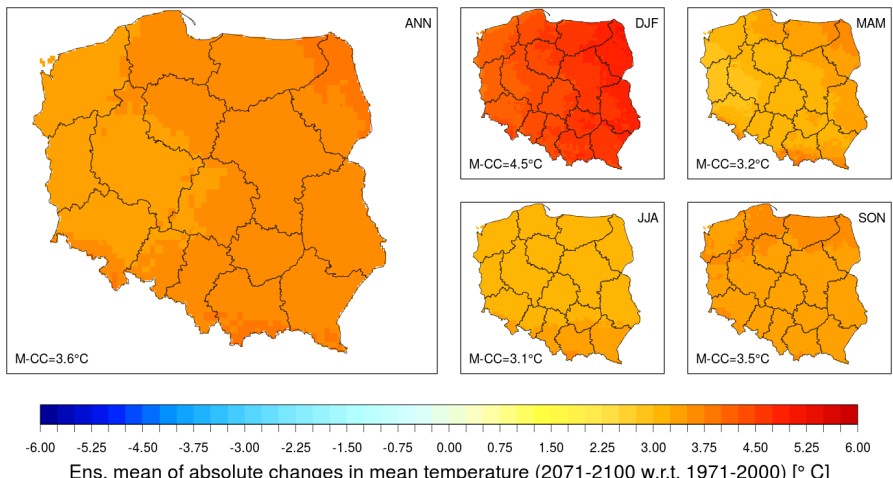

**Figure 8.** Same as Fig. 7 but for projected temperature changes (°C) in the far future (2071-2100) assuming the RCP8.5 scenarios.

pointed out a precipitation increase by up to +15% considering ten small catchments spread across the country and not all of Poland.

Assuming the RCP8.5 scenario, the expected change by the far future (2071-2100) is approximately +16% for the annual mean precipitation, with stronger increases in winter (+27%) and spring (+26%), and more moderate changes in summer (+5%) and autumn (+13%). summer exhibits almost the same change of precipitation regardless of emission scenario and time horizon.

In contrast to temperature, precipitation changes exhibit higher variability in space (spatial standard deviation averaged across all scenarios and periods equals 5%). In southern Poland, north of the Carpathian mountains, summer and autumn precipitation are even expected to decrease by as much as 5%. The influenced area is more pronounced in projections for the far future (2071-2100) and assuming the high emission scenario RCP8.5 (see Figures 9 and 10).

### 4.2 Changes in individual model simulations

### 4.2.1 Projected temperature

Results based on the individual model simulations also showed a systematic increase in both minimum and maximum temperatures for the two future periods and RCPs.

Assuming the RCP4.5 scenario, the absolute changes in annual means of minimum temperature vary between 0.8 °C and 1.6 °C (Fig. SM112). Although changes in maximum temperature are expected to have similar magnitude, changes in maximum temperature are less pronounced than for minimum temperature. On a seasonal basis, the warming is more intensified in winter



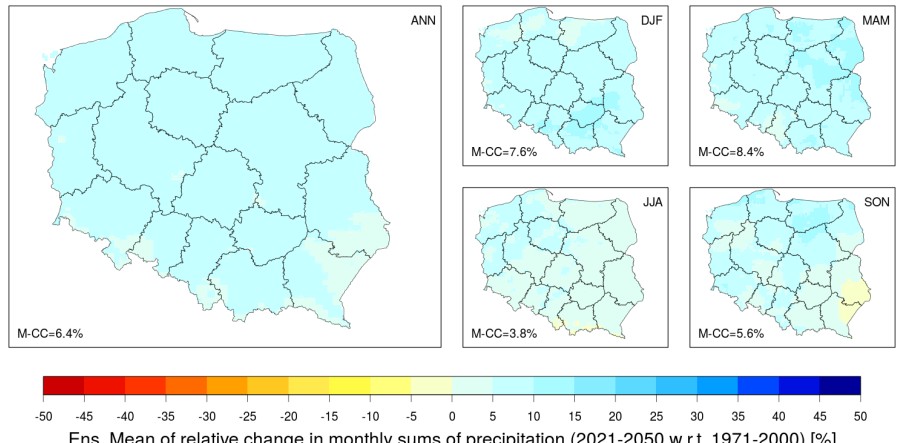

**Figure 9.** Projected changes in monthly sums of precipitation (%) for the period (2021–2050) assuming the RCP4.5 scenario. Maps show annual (left large panel) and seasonal (right small panels) changes in the multi-model ensemble mean of absolute temperature with regards to the control period (1971-2000). The legend 'M-CC' means the areal mean change estimated from the gridded data. but for projected precipitation changes (%) by 2021–2050 assuming the RCP4.5 scenario.

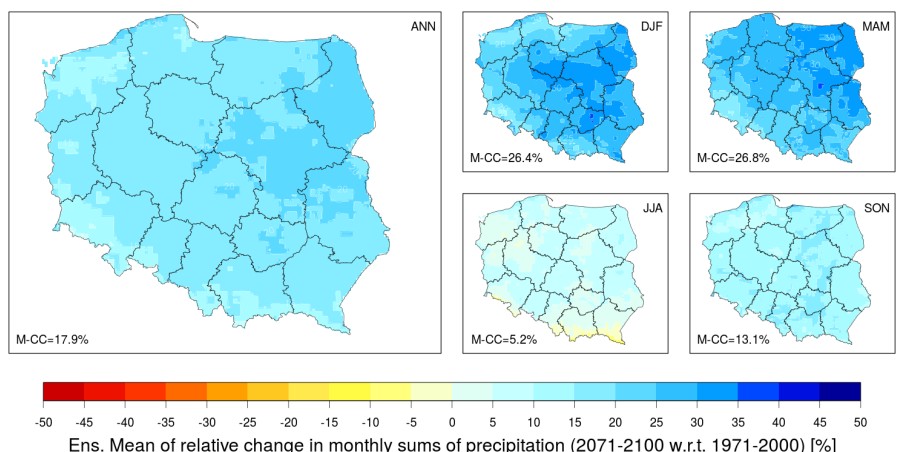

**Figure 10.** Same as Fig. 9 but for projected precipitation changes (%) by 2071–2100 assuming the RCP8.5 scenario.

(Fig. 14) and spring (Fig. SM 120) and vary from 0.42 °C (CCLM4-8-17/MPI-ESM-LR) to 2.2 °C (HIRAM5/EC-EARTH) and from 0.68 °C to 1.9 °C, respectively. The same tendency was found for maximum temperature but the magnitude of the





change is slightly amplified in spring (Fig. SM 87) which vary between 0.4 °C and 1.9 °C). The climate change in summer (Fig. SM 91) is lower than what is obtained on annual basis and the changes in absolute values vary from 0.6 °C up to 1.39 °C (Fig. SM 79). Projected minimum temperatures (Fig. 12) to the end of the 21st century are also expected to increase and vary from 1.3 to 3.7 °C in winter, 1.5 to 2.5 °C in autumn, 1.4 to 2.5 °C in summer, and 1.3 to 3.1 °C in spring. On an annual basis, the range of the changes is predicted to vary from 1.6 to 2.6 °C. Likewise, the absolute change in maximum temperature (Fig. SM 80)ranges from 1.25 and 2.7 in autumn, from 1.0 to 2.4 in summer, from 0.9 to 2.9 °C in spring, and from 1.2 to 3.2 °C in winter. On annual basis, the changes are expected to vary between 1.2 and 2.6 °C. Similarly to precipitation, both temperature variables show large differences in the projections during spring.

When assuming the RCP8.5 scenario , both minimum and maximum temperature increases show similar order of magnitude as what expected in the near future time period for all seasons and on annual scale (1 °C). However, this increase was more intensified by at least 1.2 °by the end of the 21st century and the highest warming rate is expected to occur in winter with a warming rate of  2 °C (Figures SM 81-82 and SM 114-115)).

Piniewski et al. (2017a) assessed the robustness and uncertainty of the temperature change signal using the same set of regional climate model simulations and demonstrated that the increase obtained in the annual means of daily minimum and maximum temperature was robust, however, a lower robustness was pointed out in the seasonal increases of daily minimum and maximum temperatures.

### 4.2.2 Projected precipitation

Assuming the RCP4.5 scenario, the changes in annual means of monthly sums of precipitation are projected to increase by 4 % to  11 % all over the country for the period 2021–2050. Although all models showed agreement on the overall positive change, they disagree on the spatial distribution where patches referring to slight decreases in precipitation were also found in the projections exhibiting a decreasing rate with less than 2 %, mostly located in mountainous areas. The highest increase is given by the RACMO22E model driven by the EC-EARTH model in the eastern parts of the region. On a seasonal basis, different tendencies of climate change signal were found. For winter (Fig. 13), although the overall picture of the changes suggested wetter conditions, models disagree on both the sign and magnitude, and especially, when the spatial distribution of the change is of interest. For instance, the CCLM4-8-17/CNRM-CM5 model show a dry pattern in the north eastern parts which is down to 10 % and a wet pattern in the south western parts, more pronounced in the mountainous areas where the increase can be up to 20 %. The tendency is reversed in winter precipitation modelled by CCLM4-8-17/MPI-ESM-LR global climate model, where a clear north-west to south-east gradient was found. The highest change was simulated by the regional climate model RCA4 driven by EC-EARTH model, showing an increase by up to 13 %. For summer precipitation (Fig. 15), models showed a disagreement in projecting the sign of the climate change signal, that ranged from -5% to +9%. The RCA4/MPI-ESM-LR model showed a dry pattern in southern parts of the country including the mountainous areas and can be down to 20 %, however, RACMO22E/EC-EARTH model showed an opposite tendency, albeit, the north-west to south-East gradient is reproduced. For the spring season (Fig. SM 67), there is a dominance of mostly wet patterns where precipitation changes vary from +1 % (CCLM4-8-17/CNRM-CM5 simulation) to +17 % (CCLM4-8-7/MPI-ESM-LR simulation). For the autumn



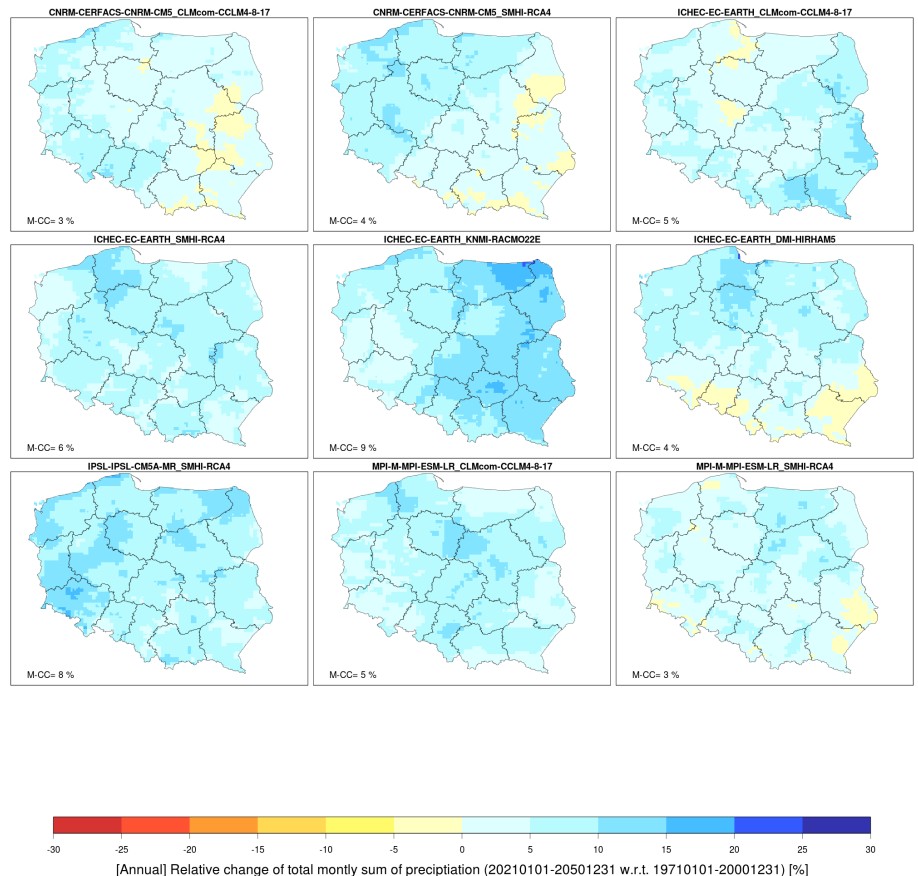

**Figure 11.** Changes in projected annual means of monthly sums of precipitation by 2021–2050 assuming the RCP4.5 scenario. The maps show relative changes with regards to the historical time period (1971–2000) for the nine bias adjusted simulations.

(Fig. 16), the relative changes are expected to vary between -4 % and 13 % and showed similar patterns to those obtained for winter, hence, no agreements between the corrected simulations was seen for the near future. The direction of the change signal became more clear when considering the end of the century and showed an overall increase on an annual scale from 4 % (RCA4/MPI-ESM-LR simulation) to 13 % (RCA4/CNRM-CM5 simulation) (Fig. SM 60). Surprisingly, on a seasonal basis,

5 all models agree on projected wet winters and springs showing the largest increase by as much as 25 % (CCLM4-8-17/EC-EARTH simulation) overall the region. Summer precipitation was found to be uncertain and is expected to vary from -8 % and 11 %. Even though, models' disagreements were more dominant in autumn and summer, it should be noted that the lowest changes were found in summer when the regional climate model was driven by the MPI-ESM-LR global climate boundaries. Contrary, the highest changes were simulated by the CCLM4-18-17 regional climate model driven by the EC-EARTH global

10 climate model and are more robust in spring than in winter.



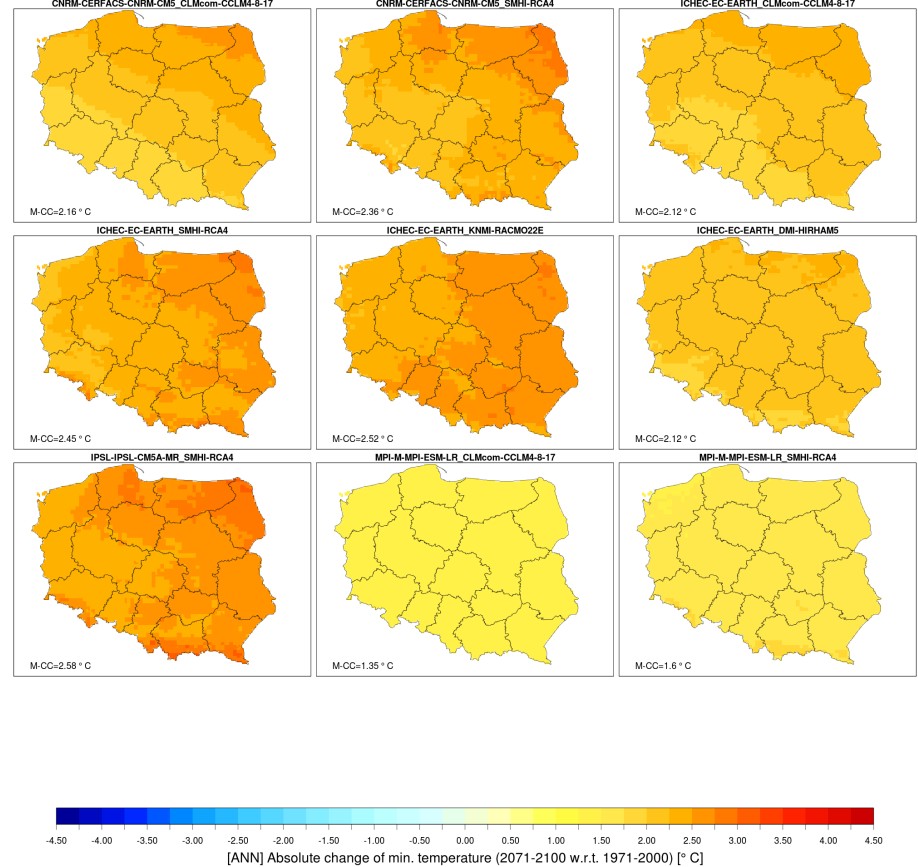

**Figure 12.** Changes in projected annual means of daily minimum temperature by 2071–2050 assuming the RCP4.5 scenario. The maps show absolute changes with regards to the historical time period (1971–2000) for the nine bias adjusted simulations.

Assuming the RCP8.5 scenario, the increase in the annual precipitation pattern became more dominant and almost all individual models projected an increase varying from +4 % to 22 % for the period 2071-2100 except the HIRHAM5/EC-EARTH model which predicted a decrease in the south-western areas but less than 5 %. This intensification could be attributed to an increase in water vapour associated with a warmer future climate conditions. However, seasonal changes showed again a
5  large disagreement between the individual models in simulating both the magnitude and the direction for the near future. For spring, all models agreed on the sign and disagreed on the magnitude. For instance, the RCA4 regional climate model driven by the EC-EARTH model projected a decrease in summer precipitation down to 6 %, while, the CCLMcom-CCLM4-8-17 regional climate model driven by CNRM-CM5 global climate model showed overall wet patterns and an increase by up to 15 % (Fig. 17). The largest models' increase was found in the winter and autumn ranging and can reach up to 33 % projected by
10  the RACMO22E/EC-EARTH model to the far future. The individual model simulations show a strong disagreement



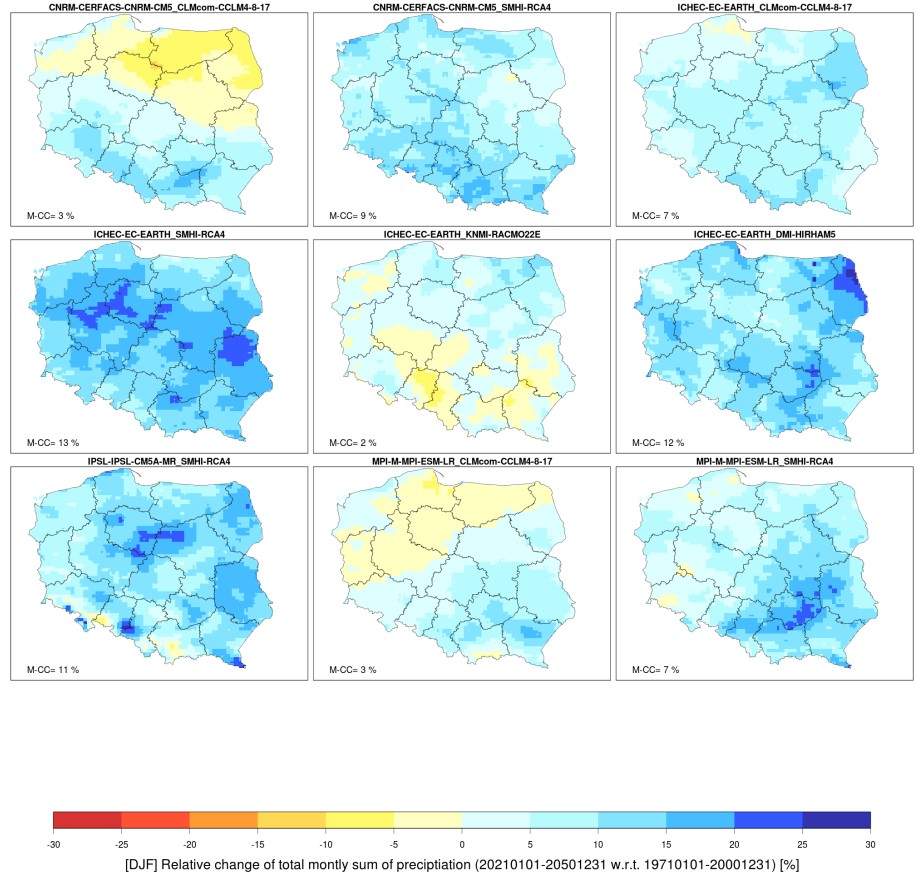

**Figure 13.** Changes in projected winter means of monthly sums of precipitation by all model simulations. The maps show the relative changes with regards to the historical period (1971–2000) for all nine adjusted simulations.

Piniewski et al. (2017a) assessed in a separate analysis the robustness of those projections and pointed out that even though the models agreed well on a precipitation increase, the changes were, in general, uncertain and not robust. They also pointed out that the spatial variability of the climate change signal was quite variable between individual climate model simulations which reduced considerably the robustness, especially for the far future.

## 5   Data availability

The CHASE-PL Climate Projection (CPLCP) dataset produced here was made available for use in two different ways: (1) in a long-lasting research data repository and (2) through a dedicated CHASE-PL web Geo-portal. The first option (Sect. 5.1) is meant to serve mainly researchers, particularly users of environmental models to apply the bias-corrected high resolution climate data as a consistent forcing dataset for projecting climate change impacts on different sectors in Poland. In this case, to





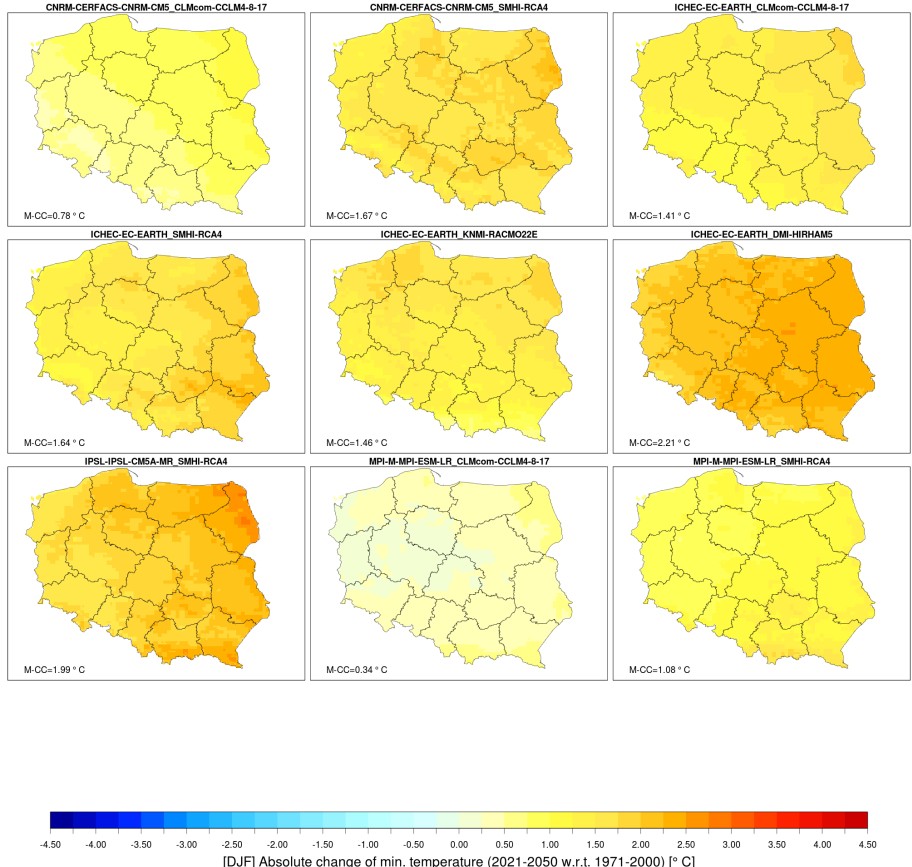

**Figure 14.** Changes in projected annual means of daily minimum temperature by 2021–2050 assuming the RCP4.5 scenario. The maps show the absolute changes with regards to the historical period (1971–2000) for all nine adjusted simulations.

achieve full consistency, it is recommended to use the observational (CPLFD-GDPT5) dataset (Berezowski et al., 2016), used as a reference for model calibration and validation. The second option (Sect. 5.2) is expected to serve both researchers and a wider audience, including students, stakeholders and public authorities, as climate change science has not been disseminated widely in Poland to date (Kundzewicz and Matczak, 2012).

## 5.1 Data repository at 4TU.Centre for Research Data

The bias-adjusted files were stored in Netcdf4 format and compiled using the Climate and Forecast (CF) conventions. The data were made available at the 4TU.Centre for Research Data (Mezghani et al., 2016). The files consisted of nine bias-adjusted regional climate model simulations of daily (minimum and maximum) temperature and precipitation for a spatial domain covering the union of Poland and the Vistula and Odra basins for one historical and two future time periods assuming the





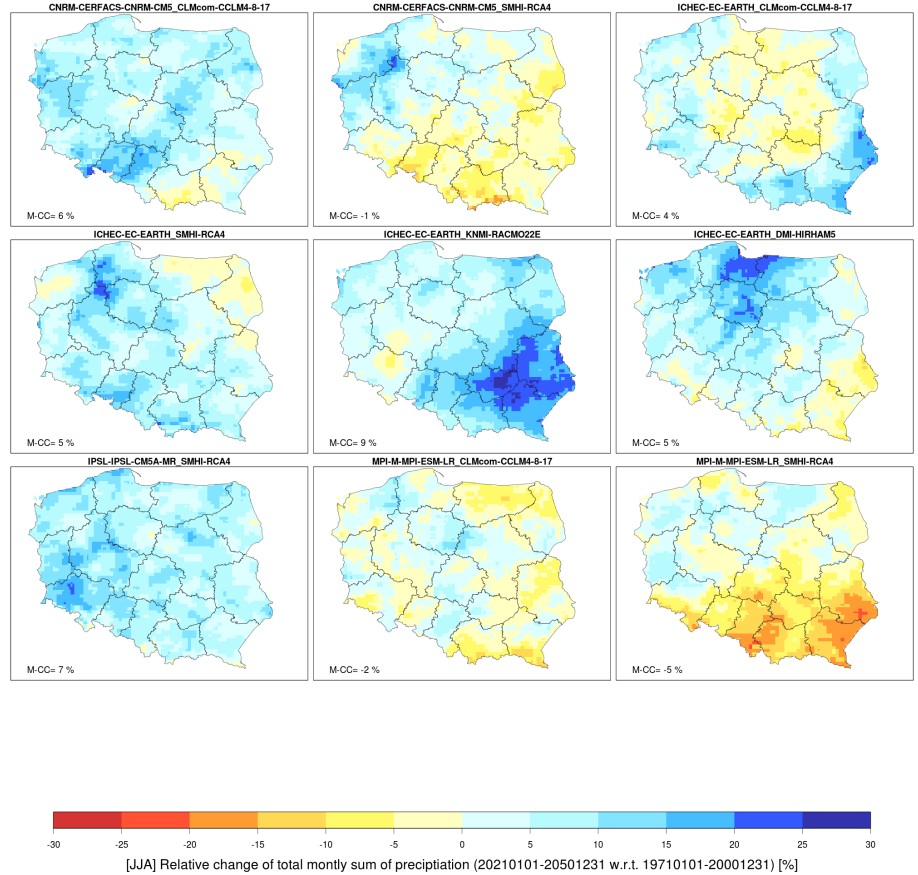

**Figure 15.** Changes in projected summer means of monthly sums of precipitation by 2021–2050 assuming the RCP4.5 scenario. The maps show the relative changes with regards to the historical period (1971–2000) for the nine adjusted simulations.

RCP4.5 and RCP8.5 scenarios. There are 135 files and the total size is 127GB. Nevertheless, the full dataset covering the continuous time period (i.e. 1950–2100) can be obtained upon request to the Norwegian Meteorological Institute. CPLCP-GDPT5 dataset is publicly available at http://dx.doi.org/10.4121/uuid:e940ec1a-71a0-449e-bbe3-29217f2ba31d.

## 5.2 Access through the Climate Impact Geoportal

5     The Climate Impact geoportal (http:\ClimateImpact.sggw.pl) developed within the CHASE-PL project presents spatial inter-active data on three aspects of climate change in Poland: (1) observations, (2) projections, and (3) impacts. The 'Observations' sub-page presents, among other things, the 5 km resolution gridded precipitation and temperature dataset CPLFD-GDPT5 (Berezowski et al., 2016) that was used in this study as the reference dataset, aggregated to monthly/seasonal/annual time series and long-term average values. The Impacts sub-page presents maps of climate change impacts on water resources (Piniewski



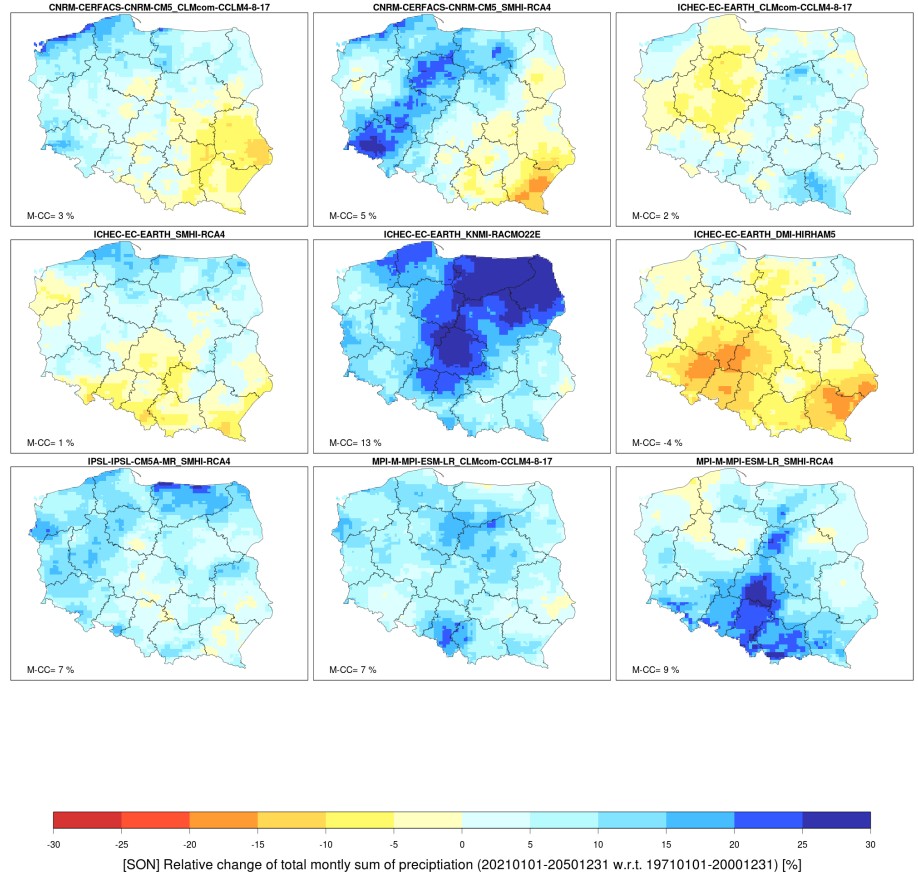

**Figure 16.** Changes in projected autumn means of monthly sums of precipitation by 2021–2050 assuming the RCP4.5 scenario. Relative changes between historical (1971-2000) and future (2020–2050) for all bias adjusted simulations assuming the RCP4.5.

et al., 2017b) obtained from hydrological modelling using SWAT driven by the dataset described in this paper. In this section we focus on the Projections sub-page presenting the contents of the CHASE-PL Climate Projections dataset (Fig. 18).

The web-map application was developed using ArcGIS Server that makes the data available using REST architecture, as well as using temporal data visualisation portal using the JavaScript API for communication between the client and the server. ESRI Geoportal Server was applied for meta-data management. Two language versions, English and Polish, are available. The Geoportal stores in total 180 maps of projected variables (precipitation, minimum and maximum temperature) for two time horizons (near and far future), under two RCPs (4.5 and 8.5), for five temporal aggregation levels (annual and four seasonal) and three ensemble statistics types (5th percentile, median and 95th percentile). All data are shown as original $5 \times 5$ km raster files. Projected changes are shown, as in this paper, as absolute differences between future and baseline periods for temperature, and as percent differences for precipitation. Ensemble statistics are calculated for projected changes across all ensemble members. By including three ensemble statistics, the geoportal informs end-users both about the magnitude and the spread of change





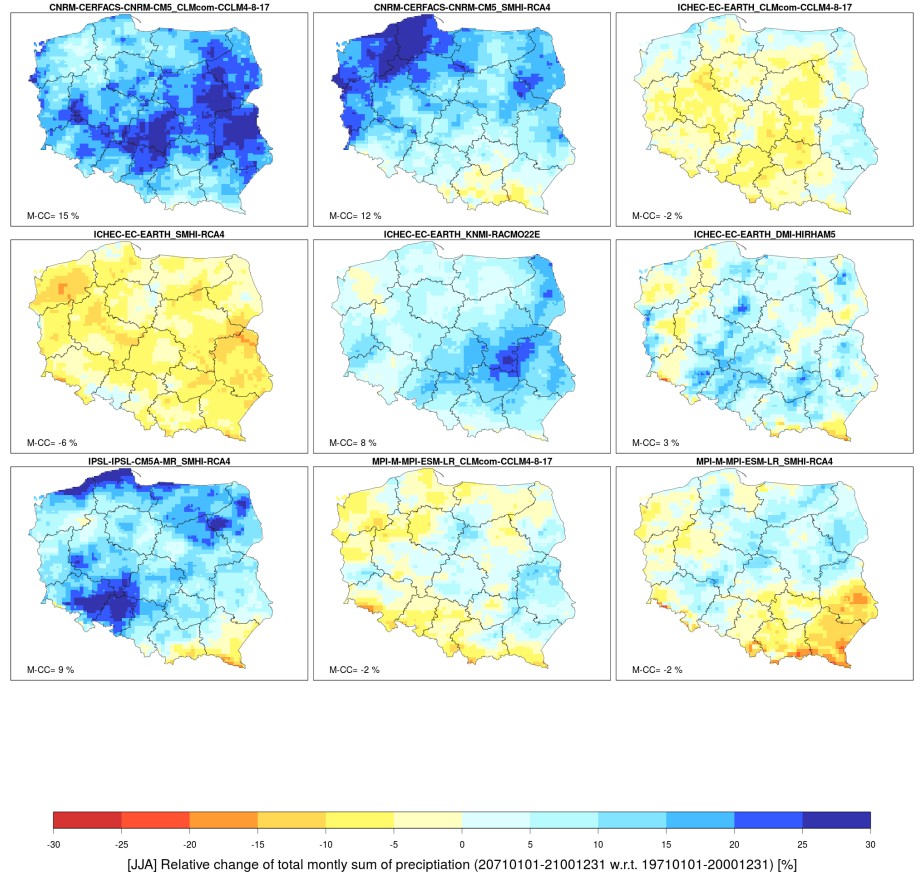

**Figure 17.** Changes in projected summer means of monthly sums of precipitation by 2071–2100 assuming the RCP8.5 scenario. The maps show the relative changes with regards to the historical period (1971–2000) for the nine adjusted simulations.

(climate model uncertainty). The web-map application has the following functionalities: (1) meta-data searching; (2) searching by location; (3) identification of selected values on the map (simultaneously for all seasons and year); (4) data download in NetCDF and GeoTIFF formats. The online help and glossary were also created in order to enhance the use of the geoportal among users less advanced in webGIS and/or climate model outputs.

## 6 Conclusions

A recent high-resolution gridded dataset was used as long term reference dataset produced over the Odra and Vistula basins in Poland and surrounding regions to adjust modelled daily precipitation and minimum and maximum temperatures by nine EURO-CORDEX regional climate models. The main purpose was to provide an update of recent climate projections for Poland assuming the new generation of radiative concentration pathways.



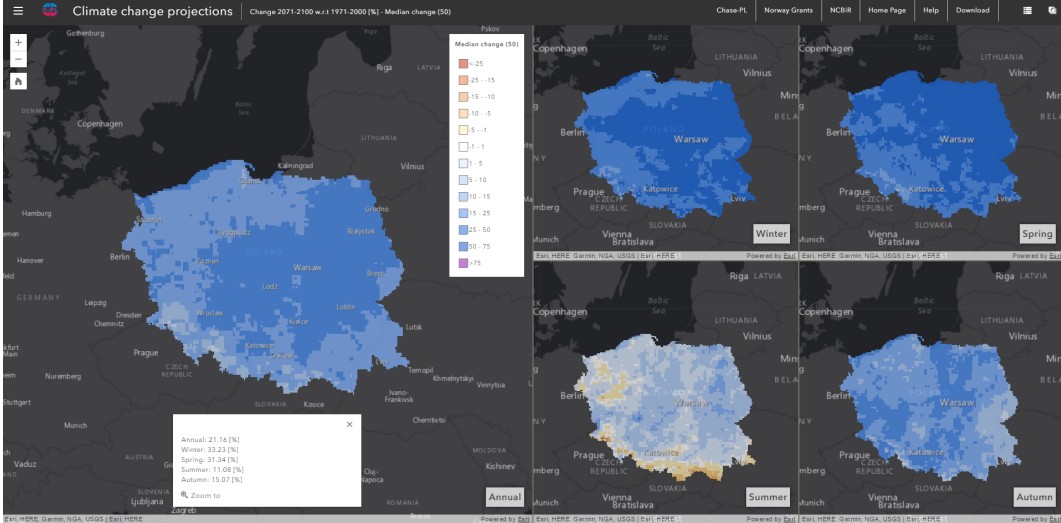

**Figure 18.** Example map of the Projections sub-page of the Climate Impact geoportal (ensemble median change in mean annual and seasonal precipitation following RCP8.5 in the far future). The white box in the bottom contains information from the Identification function on the values in selected grid point.

The bias correction was first trained on the CHASE-PL Forcing Data 'CPLFD-GDPT5' covering the time period from 1951 to 2013 and next evaluated against observations. We found that the bias correction method preserves both lower and higher order moments, including the climate change signal, hence, not affected during the bias correction process, hence, well preserved in the projections.

Once validated, the climate change signal was assessed from a multi-model ensemble of projected changes derived from a set of bias corrected EURO-CORDEX simulations. We found substantial agreement between models exhibiting an increase in both seasonal and annual means of precipitation and temperature based on the multi-model ensemble mean. We also found that this increase is projected to accelerate by the end of the 21st century and when assuming the RCP8.5 scenario. Based on the bias corrected simulations, the warming over Poland is expected to be 1 °C for the period 2021-2050 and 2 °C for the

period 2071–2100 assuming the RCP4.5 scenario. Then, it is expected to accelerate assuming the RCP8.5 from 1.3 °C for the period 2021–2050 to almost 4 °C by the end of the 21st century. Similarly to mean temperature, projected changes in regional annual means of monthly totals of precipitation are expected to increase by 6 % to 10 % and by 8 % to 16 % for the two future horizons and RCPs, respectively. This tendency was also found in projected changes in seasonal and annual means of temperature and precipitation when considering individual models' simulations rather than the multi-model ensemble mean.

However, projected changes in seasonal means of precipitation largely differ and sometimes inconsistent exhibiting spatial variations which depends on the selected season, location, future horizon and RCP. Nevertheless, the overall range of the 90% confidence interval predicted by the ensemble of multi-model simulations was found to likely vary between -7 % and +40 %,



expected to occur in summer assuming the RCP4.5 scenarios and in winter assuming the RCP8.5 scenario, respectively, at the end of the 21st century.

We believe that the CHASE-PL Climate Projection product (CPLCP-CPLFD-GDPT5) available for the period of 150 years (from 1951 to 2100) will serve as the basis for further applications, for instance, to study the impact of climate change over
Poland on many sectors (e.g. agriculture, hydrology, ecology, and tourism).

*Author contributions.*  J. E. Haugen designed the bias-adjust experiment and A. Dobler developed the model code and performed the corrections. A. Mezghani estimated the projected changes over Poland and prepared the manuscript with contributions from all co-authors.

*Competing interests.*  The authors declare that they have no conflict of interest.

*Acknowledgements.*  We acknowledge the Polish–Norwegian Research Programme operated by the National Centre for Research and Devel-
opment (NCBiR) under the Norwegian Financial Mechanism 2009–2014 to the financial support of the project CHASE-PL (Climate change impact assessment for selected sectors in Poland) in the frame of project contract no. Pol-Nor/200799/90/2014 and the World Climate Research Programme's Working Group on Regional Climate, and the Working Group on Coupled Modelling, former coordinating body of CORDEX and responsible panel for CMIP5. We also thank the climate modelling groups (listed in Table 1 of this paper) for producing and making available their model output. We also acknowledge the Earth System Grid Federation infrastructure an international effort led
by the U.S. Department of Energy's Program for Climate Model Diagnosis and Intercomparison, the European Network for Earth System Modelling and other partners in the Global Organisation for Earth System Science Portals (GO-ESSP).





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
