# Peer review of "CHASE-PL Climate Projection dataset over Poland–Bias adjustment of EURO-CORDEX simulations"

_Earth System Science Data, 2017_

## Referee Comment (RC1) · J. Wibig (Referee) · 23 Aug 2017

The paper provides an update of climate projections over Poland by adopting the new generation of concentration pathways and recent developments in climate modelling.It provides a dataset of scenarios of temperature and precipitation developed for nine individual RCM simulations and the ensamble one. For each RCM the bias is firstly assessed and than the scenario is adjusted. The scenarios are prepared on annual and seasonal resolution. Among the reasons of systematic biases in the regional climate models only two are mentioned: i) imperfect model representation of the physical processes or phenomena and ii) the parametrization or incorrect initialization

of the models. There is no suggestion that there are other causes of bias. However, the high grid resolution leads to imperfect topography and land use data. Topography is flattened and less variable affecting considerable the atmospheric circulation in particular in mountainous regions. In Poland it can considerably influence on the transport of heat and humidity in southern areas as well as the course of foehn events and spatial distribution of rainy and rain shadow areas. The other effect of coarse resolution is location of land/sea boundaries. RCM climate change projections in EURO-CORDEX are still carried out for the atmosphere only. Such models prescribe SST data from the driving GCM (Christensen et al., 2008). Consequently the quality of the prescribed SST/sea ice data depends on the quality of much coarser resolution of the global ocean component in GCM simulations. For a relatively small and semi-close seas like the Baltic Sea the effect can be quite important (Wibig et al., 2015). The discussion of these effects should be given in the introduction or in conclusions. Describing the papers on bias correction methods authors mentioned the paper by Berg et al. (2012) where three bias correction methods were applied to correct the mean and variance of precipitation and temperature modelled by the RCM COSMO-CLM driven by the ECHAM5-MPIOM GCM over entire Germany and its near surrounding areas. They stated that "the method corrects not only means but also higher moments". But it is not clear if this statement applies to all three methods or to one of them only, and if to the one only, the more detailed description of this method should be given. Further, talking over the Quantile Mapping methods based on Gutjahr and Heinemann paper (2013), the authors stated that quantile mapping method has shown a good performance in reproducing not only the mean and the standard deviation but also other statistical properties such as quantiles, as it belongs to the non-parametric family and does not require a prior knowledge of the theoretical distribution of the weather variable which makes it very attractive as it is easy to implement. However, in general the quantile mapping method may be parametric or non-parametric depending on the method of computing the quantiles. Data used in the paper are described in chapter 2. There are two data sets. First one, Polish high resolution observational climate dataset, consists

of 5 km x 5 km gridded daily precipitation, maximum and minimum temperature data. The second one consists of regional climate model simulations provided within the EURO-CORDEX initiative. As the simulation were available on different spatial resolution they were interpolated to the same grid as observational data. More details about the interpolation method should be given here. In particular some information on topography should be given and how the simulated data were altitude-adjusted to the topography of observational data. In description of the bias correction method the way of the adjustment of wet-day frequencies for precipitation should be described in details. It should be also mentioned if the quantiles for precipitation were calculated basing on all days or on wet days only, and if on wet days only the threshold for wet day should be given. In the chapter 3.2 the presentation of method is incorrect. In the description of equation 3 instead of "where s and s refer to simulated and observed values", should be "where s and o refer to simulated and observed values". In the definition of RMSE (equation 4) the quantity under root square should be divided by the number of grid points (Ng) as it is a root mean square error: . Usually when bias correction method is used, the absolute differences are used for temperature data and relative difference in case of precipitation to avoid the negative precipitation totals as a corrected values (van Roosmalen et al., 2011). Here it seems that in both cases the absolute differences were used. Why? How the problem of negative values of precipitation was solved? The other issue is the number of quantiles. The autors used 1000 uniformly distributed quantiles. In case of tails of distribution the change in correction factors can be very high from one quantile to the other, in particular for precipitation. How did you solve this problem? In further analysis the relative changes of precipitation between historical and future periods were applied. There is a clear inconsistency. Description of results is very well prepared, however some shortcomings also can be observed. When the projected temperatures are considered the increase of the annual mean temperature over Poland is presented for two periods separately, but in the case of seasonal values only one value is given for each season without information if they concern near or far future. I have also

objections to scales of some figures. For example the scale range of Fig. 7-8 are so large that the spatial differences disappear. On fig 13 the information on RCP scenario (4.5 or 8.5) should be added. On all maps the boundaries of Poland should be added. Supplementary material is very rich. Contains almost 200 maps, but there is a lot of mistakes: some maps have wrong titles, the others are located in wrong places, some are doubled. I am giving only selected examples, but because of huge number of maps I am not able to check all of them. Fig. SM61 and SM99 : figures have exactly the same titles but they are different Fig. SM60 and SM154 : figures have exactly the same titles but they are different Fig. SM59 and SM153 : figures have exactly the same titles but they are different Fig. SM62 and SM99 are identical subchapter 5.2.7 titled Projected precipitation is located in chapter 5.2 titled Projected maximum temperature. Generally my opinion on the paper is very positive. I am seeing some drawbacks presented above, however it is a first so robust and ambitious set of climate projection for Poland and the way of dissemination is very clear and easy to use. I am convinced that the shortcomings are easy to correct. References: Christensen JH, Boberg F, Christensen OB, Lucas-Picher P, 2008, On the need for bias correction of regional climate change projections of temperature and precipitation. Geohys res Lett 35:L.20709, doi:10.1029/2008GL035694. van Roosmalen L, Sonnenborg TO, Jensen KH, Christensen JH, 2011, Comparison of hydrological simulations of climate change using perturbation of observation and distribution-based scaling. Vadose J 10:136-150, doi:10.2136/vzj2010.0112 Wibig J, Maraun D, Benestad R, Kjellström E, Lorenz P, Christensen OB, 2015, Projected changes – Models and methodology, [in:] BACC II author team, Second assessment of climate change for the Baltic Sea Basin, Regional Climate Studies, Spronger Open, pp. 189-215.

Please also note the supplement to this comment:
https://www.earth-syst-sci-data-discuss.net/essd-2017-51/essd-2017-51-RC1-supplement.pdf

---

## Author Comment (AC1) · 15 Sep 2017

J. Wibig (Referee)
zameteo@uni.lodz.pl

*We would like to thank the reviewer for her positive opinion regarding our paper. Hereafter, we address a point by point answer. Referee text is in grey and answers are in black justified text.*

Generally my opinion on the paper is very positive. I am seeing some drawbacks presented above, however it is a first so robust and ambitious set of climate projection for Poland and the way of dissemination is very clear and easy to use. I am convinced that the shortcomings are easy to correct.

> Indeed, the main purpose was to provide a robust update of climate projections for Poland which will serve as basis for many sectors, especially, those directly affected by a climate change.

The paper provides an update of climate projections over Poland by adopting the new generation of concentration pathways and recent developments in climate modelling.It provides a dataset of scenarios of temperature and precipitation developed for nine individual RCM simulations and the ensemble one. For each RCM the bias is firstly assessed and then the scenario is adjusted. The scenarios are prepared on annual and seasonal resolution.

Among the reasons of systematic biases in the regional climate models only two are mentioned: i) imperfect model representation of the physical processes or phenomena and ii) the parametrization or incorrect initialization of the models. There is no suggestion that there are other causes of bias. However, the high grid resolution leads to imperfect topography and land use data. Topography is flattened and less variable affecting considerable the atmospheric circulation in particular in mountainous regions. In Poland it can considerably influence on the transport of heat and humidity in southern areas as well as the course of foehn events and spatial distribution of rainy and rain shadow areas. The other effect of coarse resolution is location of land/sea boundaries. RCM climate change projections in EURO-CORDEX are still carried out for the atmosphere only. Such models prescribe SST data from the driving GCM (Christensen et al., 2008). Consequently the quality of the prescribed SST/sea ice data depends on the quality of much coarser resolution of the global ocean component in GCM simulations. For a relatively small and semi-close seas like the Baltic Sea the effect can be quite important (Wibig et al., 2015). The discussion of these effects should be given in the introduction or in conclusions.

In our opinion, the influence of the `topography` and the `land/sea boundaries` is included in one of the two reasons related to systematic biases that we mentioned in the manuscript, i.e.*"imperfect model representation of the physical processes or phenomena"* which holds for both GCMs and RCMs. Also, discussing this issue in the introduction would mislead the scope of the paper. However, we added details related to this issue in the conclusions and referred to Wibig et al. (2015) study.

*`The high-resolution (5 x 5 km) gridded observational data set (CPLFD-GDPT5; Berezowski et al. 2016) was used to train the bias correction method, and as the former included a correction of the altitudinal effects for both temperature and precipitation, both northern and southern (mountains) topographical features were, indirectly, taken into account during the correction procedure. That is why we haven't performed any altitudinal correction in the regridding of each RCM. Regarding the climate projections, we demonstrated that the climate change signal was not affected by the bias correction method. Yet, any misrepresentation of the climate change signal in the regional climate model, due to inherited misrepresentation of the SSTs and Sea/Ice extent, especially, in the northern part of the Baltic Sea and the mountains in the southern parts, could have an influence on the projected temperature and precipitation changes. In this case, we assumed that the changes are less affected, thus, more robust than the absolute values.`*

*Describing the papers on bias correction methods authors mentioned the paper by Berg et al. (2012) where three bias correction methods were applied to correct the mean and variance of precipitation and temperature modelled by the RCM COSMO-CLM driven by the ECHAM5-MPIOM GCM over entire Germany and its near surrounding areas. They stated that " the methods corrects not only means but also higher moments". But it is not clear if this statement applies to all three methods or to one of them only, and if to the one only, the more detailed description of this method should be given.*

We fully agree that the description of the paper by Berg et al. (2012) was unclear. In their paper, Berg et al. (2012) reported that the success of the correction method is related to the magnitude of the bias. They also pointed out that for low biases all methods are able to reproduce the mean and variance, however, if the model bias is high, *"different adverse effects are to be expected"* in reproducing, especially, higher order moments. To avoid any confusion, we added the word "some" before "the methods …".

*Further, talking over the Quantile Mapping methods based on Gutjahr and Heinemann paper (2013), the authors stated that quantile mapping method has shown a good performance in reproducing not only the mean and the standard deviation but also other statistical properties such as quantiles, as it belongs to the non-parametric family and does not require a prior knowledge of the theoretical distribution of the weather variable which makes it very attractive as it is easy to implement. However, in general the quantile mapping method may be parametric or non-parametric depending on the method of computing the quantiles.*

We agree that the quantile mapping can be parametric or nonparametric. We modified the text and also replaced the reference "Gutjahr and Heinemann (2013)" with "Fang et al. (2015)" as we think it is more appropriate. In the paper of Fang et al. (2015), the quantile mapping is defined as a non parametric bias correction method. But we agree that this could be a bit confusing.

The new text states :
*"Among existing methods, the nonparametric quantile mapping method, referred later to as the `Quantile Mapping (QM)` for simplicity, has shown a good performance in reproducing not only the mean and the standard deviation but also other statistical properties such as quantiles (Fang et al., 2015)."*

Data used in the paper are described in chapter 2. There are two data sets. First one, Polish high resolution observational climate dataset, consists of 5 km x 5 km gridded daily precipitation, maximum and minimum temperature data. The second one consists of regional climate model simulations provided within the EURO-CORDEX initiative. As the simulation were available on different spatial resolution they were interpolated to the same grid as observational data. More details about the interpolation method should be given here. In particular some information on topography should be given and how the simulated data were altitude-adjusted to the topography of observational data.

We used the nearest neighbor interpolation method which means that each cell in the new (in this case the 5 x 5 km high- resolution) was assigned the RCM values of the nearest cell in their original grid resolution. Moreover, we did not correct the interpolated values to altitudinal variations as it was included in the observational grid dataset. Thus, we did not include any correction in the bias adjustment procedure. The main reason is that this step was unnecessary as projected climate change signal was not affected by such a correction when expressed in terms of absolute values for temperature and relative differences for precipitation with regards to a reference value.

In description of the bias correction method the way of the adjustment of wet-day frequencies for precipitation should be described in details.

An adjustment of the wet-day frequency was also performed as modelled values corresponding to the dry part of the observed empirical CDF were set to zero. First, the frequency of wet-days is derived from observations and the simulated value that corresponds to the same frequency in the simulations is used as threshold. Then, all modelled values below this threshold are set to zero. This ensures equal fraction of days with precipitation in the observed and the modelled data. The transformations are only fitted to the portion of the distributions corresponding to observed wet days. Accordingly, we modified the manuscript to include these details.

It should be also mentioned if the quantiles for precipitation were calculated basing on all days or on wet days only, and if on wet days only the threshold for wet day should be given.

> The function F in equation 1 was fitted to the fraction of observed wet days. A wet-day is defined as a day with a precipitation amount higher than 0 mm/day.

In the chapter 3.2 the presentation of  is incorrect. In the description of equation 3 instead of "where s and s refer to simulated and values", should be "where s and o refer to simulated and observed values".

> Corrected

In the definition of RMSE (equation 4) the quantity under root square should be divided by the number of grid points (Ng) as it is a root mean square error:

> Corrected

Usually when bias correction method is used, the absolute differences are used for temperature data and relative difference in case of precipitation to avoid the negative precipitation totals as a corrected values (van Roosmalen et al., 2011). Here it seems that in both cases the absolute differences were used. Why? How the problem of negative values of precipitation was solved?

> We do not agree with the referee here. To which negative values of precipitation the reviewer is referring to? In our opinion, the quantile mapping method should not lead to negative corrected values of modeled precipitation as all quantiles are retrieve from the distribution of the observed wet-day values and will always be positive. We did not subtract any value when the correction is made.

The other issue is the number of quantiles. The authors used 1000 uniformly distributed quantiles. In case of tails of distribution the change in correction factors can be very high from one quantile to the other, in particular for precipitation. How did you solve this problem?

> Yes, we totally agree with the reviewer and this explains why we used a large number of quantiles. This number is usually set to 101 and we intentionally increased it to 1000 to minimize the errors between the distribution of observed and corrected values. Please also note that the quantile mapping was done on seasonal basis, thus 1000 quantiles is in the same order of magnitude as the total number of wet-days in each training period (e.g. ~30 yrs * ~30 wet-days/season = 900 data points). We hope this helped understanding this point.

In further analysis the relative changes of precipitation between historical and future periods were applied. There is a clear inconsistency.

As mentioned above, we used absolute values of the bias (simulated - observed) to evaluate the model biases in both temperature and precipitation. For the climate projections, we used absolute changes in temperature and relative changes in precipitation. We do not see any inconsistency here as the main messages behind those analyses is different.

Description of results is very well prepared, however some shortcomings also can be. When the projected temperatures are considered the increase of the annual mean temperature over Poland is presented for two periods separately, but in the case of seasonal values only one value is given for each season without information if they concern near or far future.

The first paragraph discussed the projected results assuming the RCP4.5 scenario and the second paragraph is devoted to RCP 8.5 projected results. We modified the beginning of the first paragraph starting with `Assuming the RCP4.5,...` to avoid any confusion.

I have also objections to scales of some figures. For example the scale range of Fig. 7-8 are so large that the spatial differences disappear.

We agree. However, the main purpose of using a large color bar is to cover all projected changes looking at all future time horizons and emission scenarios. A common scale of the color bar made the comparison across future periods and emission scenarios easy to interpret. In addition, the spatial variability in Figure 7 is very low assuming RCP4.5 scenario compared to the RCP8.5 scenario.

On fig 13 the information on RCP scenario (4.5 or 8.5) should be added.

Added

On all maps the boundaries of Poland should be added.

Yes, we fully agree in adding the country borders that would definitely help reading the maps.

Supplementary material is very rich. Contains almost 200 maps, but there is a lot of mistakes: some maps have wrong titles, the others are located in wrong places, some are doubled. I am giving only selected examples, but because of huge number of maps I am not able to check all of them.

Fig. SM61 and SM99 : figures have exactly the same titles but they are different
Corrected
Fig. SM60 and SM154 : figures have exactly the same titles but they are different Fig.

Corrected

SM59 and SM153 : figures have exactly the same titles but they are different

Corrected

Fig. SM62 and SM99 are identical

Corrected

subchapter 5.2.7 titled Projected precipitation is located in chapter 5.2 titled Projected maximum temperature.

Thank you for this comment. Actually, the subsection related to 'projected precipitation' was duplicated twice. We removed it.

References:
Christensen JH, Boberg F, Christensen OB, Lucas-Picher P, 2008, On the need for bias correction of regional climate change projections of temperature and precipitation. Geohys res Lett 35:L.20709, doi:10.1029/2008GL035694.

van Roosmalen L, Sonnenborg TO, Jensen KH, Christensen JH, 2011, Comparison of hydrological simulations of climate change using perturbation of observation and distribution-based scaling. Vadose J10:136-150, doi:10.2136/vzj2010.0112

Wibig J, Maraun D, Benestad R, Kjellström E, Lorenz P, Christensen OB, 2015, Projected changes – Models and methodology, [in:] BACC II author team, Second assessment of climate change for the Baltic Sea Basin, Regional Climate Studies, Spronger Open, pp. 189-215.

Please also note the supplement to this comment:
https://www.earth-syst-sci-data-discuss.net/essd-2017-51/essd-2017-51-RC1-supplement.pdf

Fang, G. H., J. Yang, Y. N. Chen, and C. Zammit. 2015. "Comparing Bias Correction Methods in Downscaling Meteorological Variables for a Hydrologic Impact Study in an Arid Area in China." Hydrol. Earth Syst. Sci. 19 (6): 2547–59. doi:10.5194/hess-19-2547-2015.

Berezowski, Tomasz, Mateusz Szcześniak, Ignacy Kardel, Robert Michałowski, Tomasz Okruszko, Abdelkader Mezghani, and Mikołaj Piniewski. 2016. "CPLFD-GDPT5: High-Resolution Gridded Daily Precipitation and Temperature Data Set for Two Largest Polish River Basins." *Earth System Science Data* 8 (1): 127–39. doi:10.5194/essd-8-127-2016.

---

## Referee Comment (RC2) · M. Miętus (Referee) · 19 Sep 2017

I welcome this manuscript with great interest and satisfaction that we have strong significant progress in respect of climate scenarios for Poland. High quality date sets presented by authors of the manuscript looks promising. Such data sets are needed and of high value. Additionally manuscript describes procedure of bias adjustment of Euro-Cordex simulation. It is significant progress. By this procedure user receive higher quality data of high resolution in space which might be use for impact assessments and adaptations strategies what is important from many point of view. The only problem I can see is related with references. Author did not mention few earlier

published papers discussing bias adjustment to future climate simulation for Poland, both by means of RCM and statistical downscaling. I also would suggest to redraw all maps in figures 1-6 because the shape of Poland is not natural in meridional direction. I would like to mentioned some problem with access to The Climate Impact geoportal (http:\ClimateImpact.sggw.pl) developed within the CHASE-PL project and to some limitation of this data set in respect of observations what is not the subject of this manuscript and my review but might have impact on results.

---

## Author Comment (AC2) · 16 Oct 2017

J. Wibig (Referee)
zameteo@uni.lodz.pl

*We would like to thank the reviewer for her positive opinion regarding our paper. Hereafter, we address a point by point answer. Referee text is in grey and answers are in black justified text.*

*We hereby address additional modifications in the manuscript as requested by the reviewer but were not mentioned in the previous reply.*

Description of results is very well prepared, however some shortcomings also can be. When the projected temperatures are considered the increase of the annual mean temperature over Poland is presented for two periods separately, but in the case of seasonal values only one value is given for each season without information if they concern near or far future.

> We also would like to withdraw the reviewer's attention that we have developed more the subsections 4.1.1 related to projected temperature changes in the multi-model ensemble mean and section 4.2 related to changes in individual model simulations as well as the conclusions.

On all maps the boundaries of Poland should be added.

> We updated all figures by adding Poland boundaries. We also improved the quality of all the maps as well as the font size of the legends.

Supplementary material is very rich. Contains almost 200 maps, but there is a lot of mistakes: some maps have wrong titles, the others are located in wrong places, some are doubled. I am giving only selected examples, but because of huge number of maps I am not able to check all of them.

> The supporting material has been substantially modified. We added more details and inserted one table giving the full list of simulations (Same as Table 1, in the main manuscript). We also reorganised the different sections, improved the quality of all the figures, and corrected all figures captions. We also added more references to the SM in the main manuscript.

> Finally, we would like to further thank the two reviewers for their constructive comments. Accordingly, we added the following statement into the acknowledgment.

*" Finally, we would like to thank the two reviewers Joanna Wibig and Mirosław Miętus for their respective positive and constructive comments that helped improving this manuscript. "*

---

## Author Comment (AC3) · 16 Oct 2017

Mirosław Miętus (Referee)
miroslaw.mietus@ug.edu.pl

*We would like to thank the reviewer for his very positive opinion regarding our paper. Hereafter, we address a point by point answers. Referee text is highlighted in grey and answers are in black justified text. The added statements into the the text are highlighted in italic.*

I welcome this manuscript with great interest and satisfaction that we have strong significant progress in respect of climate scenarios for Poland. High quality data sets presented by authors of the manuscript looks promising. Such data sets are needed and of high value. Additionally manuscript describes procedure of bias adjustment of Euro-Cordex simulation. It is significant progress. By this procedure user receive higher quality data of high resolution in space which might be use for impact assessments and adaptations strategies what is important from many point of view.

Indeed, as mentioned in the reply to the first reviewer[1], the main purpose was to provide an update of high resolution climate projections over Poland including the latest generation of emission scenarios and climate model simulations which can serve as basis for many impact assessments and adaptations strategies related to climatic changes.

The only problem I can see is related with references. Author did not mention few earlier published papers discussing bias adjustment to future climate simulation for Poland, both by means of RCM and statistical downscaling.

We have addressed this point in the introduction, specifically, in the seventh and eighth paragraphs. We intentionally dedicated these two paragraphs to highlight the small number of studies related to climate projections over Poland and not elsewhere in the manuscript (i.e. different subsections). We discussed studies related to climate projections based on 1) the ENSEMBLES project ($7) and 2) the newest generation of climate model simulations ($8) using both dynamical (e.g. bias corrected RCM simulations) and statistical downscaling methods. For instance, we discussed the main findings within the 'KLIMADA' project, where bias corrected simulations from the ENSEMBLES project were used to analyse future climatic changes. We added the reference *"Osuch et al. (2012)"* to refer to the 'KLIMADA' project. We also discussed the work of Piotrowski and Jędruszkiewicz (2013) based on the ENSEMBLES project
* * *
[1] https://editor.copernicus.org/index.php/essd-2017-51-AC1.pdf?_mdl=msover_md&_jrl=386&_lcm=oc108lcm109w&_acm=get_comm_file&_ms=59622&c=129626&salt=7757660411093808917

followed by the work of Pluntke et al. (2016). The latter was the unique study we found which was based on statistical techniques to downscale global climate model simulations over Poland. We also added the work of *Osuch et al. (2016)* dealing with bias correction of six simulations taken from the ENSEMBLES project into paragraph 8. We would be pleased to add any missing or recent work related to bias corrected climate model or statistically downscaled simulations over Poland to additionally be discussed in the manuscript. Though, we welcome the reviewer to specify any missing or recent references related to such topic.

I also would suggest to redraw all maps in figures 1-6 because the shape of Poland is not natural in meridional direction.

Based on this comment, and the comment of reviewer #1 we have remapped all figures 1 to 6 and improved substantially their quality. We hope that both the reviewers and the readers are more comfortable with the new changes.

I would like to mentioned some problem with access to The Climate Impact geoportal (ClimateImpact.sggw.pl) developed within the CHASE-PL project and to some limitation of this data set in respect to observations what is not the subject of this manuscript and my review but might have impact on results.

We have verified the access to the climate projection web portal and made sure that it is working properly now. Yet, the web portal could sometimes get down depending on the speed of the internet connection and the number of simultaneous users connected to the web portal.

Generally my opinion on the paper is very positive. I am seeing some drawbacks presented above, however it is a first so robust and ambitious set of climate projection for Poland and the way of dissemination is very clear and easy to use. I am convinced that the shortcomings are easy to correct.

We also would like to withdraw the reviewer's attention that we have developed more the subsections 4.1.1 related to projected temperature changes in the multi-model ensemble mean and section 4.2 related to changes in individual model simulations as well as the conclusions as requested by reviewer #1. The supporting material has been modified substantially where we reorganised the different sections and improved the quality of similar maps to Figures 1 to 6 based on the full set of the ensemble simulations.

Finally, we would like to further thank the two reviewers for their constructive comments. Accordingly, we added the following statement into the acknowledgment.

*" Finally, we would like to thank the two reviewers Joanna Wibig and Mirosław Miętus for their respective positive and constructive comments that helped improving this manuscript. "*

References

Osuch, M., R. J. Romanowicz, D. Lawrence, and W. K. Wong. 2016. "Trends in Projections of Standardized Precipitation Indices in a Future Climate in Poland." *Hydrol. Earth Syst. Sci.* 20 (5):1947–69. https://doi.org/10.5194/hess-20-1947-2016.

Osuch, Marzena, Kindler, R. J. Romanowicz, K. Berbeka, and A. Banrowska. 2012. "KLIMADA Strategia Adaptacji Polski Do Zmian Klimatu w Zakresie Sektora 'Zasoby i Gospodarka Wodna.'" KLIMADA project, IGF PAN, Warsaw, 245 pp.

Piotrowski, Piotr, and Joanna Jędruszkiewicz. 2013. "Projections of Thermal Conditions for Poland for Winters 2021-2050 in Relation to Atmospheric Circulation." *Meteorologische Zeitschrift*, October, 569–75. https://doi.org/10.1127/0941-2948/2013/0450.

Romanowicz, Renata J, Ewa Bogdanowicz, Sisay E Debele, Joanna Doroszkiewicz, Hege Hisdal, Deborah Lawrence, Hadush K Meresa, et al. 2016. "Climate Change Impact on Hydrological Extremes: Preliminary Results from the Polish-Norwegian Project." *Acta Geophysica* 64 (2):477–509. https://doi.org/10.1515/acgeo-2016-0009.